# Unified World Models: Memory-Augmented Planning and Foresight for Visual Navigation

## Abstract

Enabling embodied agents to effectively *imagine* future states is critical for robust and generalizable visual navigation. Current state-of-the-art approaches, however, adopt modular architectures that separate navigation planning from visual world modeling, leading to state–action misalignment and limited adaptability in novel or dynamic scenarios. To overcome this fundamental limitation, we propose **UniWM**, a unified, memory-augmented world model integrating egocentric visual foresight and planning within a single multimodal autoregressive backbone. Unlike modular frameworks, UniWM explicitly grounds action decisions in visually imagined outcomes, ensuring tight alignment between prediction and control. A hierarchical memory mechanism further integrates detailed short-term perceptual cues with longer-term trajectory context, enabling stable, coherent reasoning over extended horizons. Extensive experiments across four challenging benchmarks (Go Stanford, ReCon, SCAND, HuRoN) demonstrate that UniWM substantially improves navigation success rates by up to 30%, significantly reduces trajectory errors compared to strong baselines, and exhibits impressive zero-shot generalization on the unseen TartanDrive dataset. These results highlight UniWM as a principled step toward unified, imagination-driven embodied navigation.

## 1 Introduction

Visual navigation is a fundamental capability for embodied AI and autonomous systems Mirowski et al. (2016); Chaplot et al. (2020); Fu et al. (2022); Sridhar et al. (2024), enabling intelligent agents to interpret egocentric visual inputs and sequentially select actions toward goals within complex environments Karnan et al. (2022). This skill underlies critical real-world applications such as robotic delivery, autonomous driving, and assistive technologies, demanding robust perception, precise planning, and the capacity to *anticipate* environmental evolution resulting from potential actions. Humans naturally excel at such imaginative reasoning, routinely performing mental simulations to plan routes effectively through both familiar and novel scenarios Bar et al. (2025).

Despite rapid progress in visual navigation, existing approaches remain constrained by fundamental limitations (Figs. 1). **(a) Direct policy methods** (e.g., GNM Shah et al. (2022), VINT Shah et al. (2023), NoMaD Sridhar et al. (2024)) map observations directly to action sequences. Although effective within familiar distributions, such policies are rigidly tied to training data and fail to adapt in novel environments Song et al. (2025). **(b) Modular pipelines** seek to remedy this by coupling a planner with a separate world model: NavCoT Lin et al. (2024) textualizes future observations, inevitably discarding spatial fidelity, while NWM Bar et al. (2025) employs diffusion models to generate candidate visual rollouts that are subsequently ranked. Yet, when prediction and control are learned in isolation and trajectory memory is absent, state–action misalignment emerges, and errors accumulate under partial observability and extended horizons Ding et al. (2024); Xiao et al. (2025). **(c) Unified autoregressive frameworks** offer a more principled alternative by interleaving "imagining the next view" with "predicting the next action," grounding decisions in envisioned outcomes and thereby reducing misalignment (Fig. 1c). However, unification alone cannot halt the gradual drift inherent to longer-horizon reasoning. **(d) Hierarchical memory** provides the missing inductive bias: by retaining both immediate perceptual cues and longer-range trajectory context, it endows model with temporal coherence, yielding the highest SR and lowest errors in challenging settings (Fig. 1d). In essence, navigation demands not only ability to *imagine while acting* but also to

Figure 1: **Comparison of goal-conditioned visual navigation methods.** All panels use the same start/goal observations; headers report navigation performance SR↑, ATE↓, and RPE↓ on HuRoN Hirose et al. (2023) dataset. **(a)** Navigation policy methods like NoMaD Sridhar et al. (2024) directly predict action sequences $A_T$. **(b)** World model for navigation like NWM Bar et al. (2025) uses a world model to visualize future observations, enhancing a separate navigation planner. **(c)** UniWM (no memory) unifies planning and visualization within one multimodal backbone, and actions are grounded in the imagined next observation while generating $A_T$ autoregressively. **(d)** UniWM (with hierarchical memory) adds intra-step and cross-step memory banks, stabilizing longer-horizon rollouts and consistently yielding the highest SR and lowest errors (ATE/RPE).

*remember over time.* The central challenge, therefore, is to couple planning and imagination within a unified backbone while embedding temporal structure to ensure stable longer-horizon performance.

In response, we propose **UniWM**, a unified memory-augmented world model that integrates navigation planning and visual imagination within a single multimodal autoregressive backbone (Fig. 2; Sec. 2.1). During training we interleave planner and world-model samples and jointly optimize bin-token classification for actions and reconstruction for images in a shared tokenization space for actions, text, pose, and vision; the framework scales with parameter-efficient fine-tuning such as LoRA (Fig. 2 (a); Sec. 2.2). At inference UniWM alternates between predicting the next action and imagining the next egocentric view, which explicitly grounds control in predicted visual outcomes and mitigates state and action misalignment (Fig. 2 (b); Sec. 2.3). Additionally, a hierarchical two-level memory that combines an intra-step cache with a cross-step trajectory store augments attention through similarity gating and temporal decay, sustaining coherent longer-horizon rollouts and improving stability (Fig. 3). Together these components unify planning and imagination within one backbone and provide a practical recipe for memory-augmented foresight in visual navigation.

Empirically, UniWM improves Success Rate and reduces ATE and RPE across Go Stanford, ReCon, SCAND, and HuRoN relative to GNM, VINT, NoMaD, Anole-7B, and NWM. For example, on Go Stanford the SR increases from 0.45 to 0.75 (Table 1; Fig. 4). On the unseen TartanDrive, UniWM generalizes without fine-tuning and attains an SR of 0.42 (Table 8; Fig. 6). UniWM also delivers stronger one-step and rollout visualization quality, with higher SSIM and PSNR and lower LPIPS and DreamSim (Table 2). Ablation studies clarify the sources of improvement: reconstruction enhances imagination fidelity and indirectly aids navigation; the bin-token loss directly improves action accuracy; hierarchical memory is essential for longer-horizon stability; and token budget, memory layer selection, and substep interleaving explain the remaining gains (Tables 3, 5, 6; Fig. 5).

In summary, this paper provides the following key contributions:

- **Unified architecture.** We propose UniWM, the first unified, memory-augmented world model integrating visual navigation planning and imagination within a single multimodal autoregressive backbone, effectively addressing representational fragmentation inherent in modular approaches.

- **Unified training.** We propose an end-to-end interleaved training strategy that unifies planner and world-model instances within a single autoregressive backbone, jointly optimizing discretized action prediction and visual reconstruction to tightly align imagination with control.

- **Hierarchical memory.** We introduce a hierarchical memory mechanism that fuses short-term perceptual details and longer-term trajectory context through similarity-based retrieval and temporal weighting, enabling stable and coherent predictions over extended navigation horizons.

- **Comprehensive validation.** Extensive experiments validate UniWM's significant improvement over state-of-the-art methods across multiple benchmarks, demonstrating superior imagination fidelity, enhanced navigation performance, and robust generalization to novel scenarios.

## 2 UNIWM FRAMEWORK

We present **UniWM**, a unified, memory-augmented world model that performs *planning* and *visualization* within an autoregressive multimodal backbone. We first introduce navigation preliminaries and a unified formulation that replaces the disjoint planner–world-model pair with one multimodal LLM augmented by hierarchical memory (Eq.2, Sec.2.1). We then describe the unified training scheme, *i.e.* multimodal tokenization and role-specific objectives for planning and world modeling (Sec.2.2), and hierarchical memory for stable longer-horizon rollouts at inference (Sec.2.3).

### 2.1 PRELIMINARIES AND UNIFIED FORMULATION

Given an egocentric RGB observation $o_s$ at the start, the initial agent pose $p_0 \in \mathbb{R}^3$ (position and yaw), and the goal observation $o_g$, the agent must predict a sequence of navigation actions $A_T = \{\hat{a}_1, \hat{a}_2, \ldots, \hat{a}_T\}$ that leads to the goal Sridhar et al. (2024). Each action $\hat{a}_t$ is either a continuous control command $(\mathbf{u}_t, \phi_t)$ or a terminal Stop, where $\mathbf{u}_t \in \mathbb{R}^2$ encodes planar translation (forward/backward, left/right) and $\phi_t \in \mathbb{R}$ encodes yaw rotation Bar et al. (2025). Actions are executed sequentially, and agent is required to make monotonic progress toward $o_g$ until issuing Stop.

**World Models for Navigation.** World models Ha & Schmidhuber (2018) predict future environment states (often represented as image frames or video segments), conditioned on the current state and conditional variables. Formally, this can be written as $\hat{s}_{t+1} = \mathcal{W}(\hat{s}_t, \mathbf{c})$, where $\hat{s}_t$ is the current state, $\hat{s}_{t+1}$ the predicted next state, and $\mathcal{W}$ the learned world model. The conditioning context $\mathbf{c}$ may include the executed action $a_t$, natural-language instructions, history of past observations, or other environmental factors Russell et al. (2025). In navigation, world models serve as imagination engines that anticipate future observations to support action planning. This typically involves two coupled modules Bar et al. (2025): a **planner**, which selects next action given current observation and goal; and a **world model**, which simulates the consequent observation conditioned on the chosen action and contextual cues such as start and goal views. Their interaction can be formalized as:
$$\hat{a}_{t+1} = \mathcal{P}(\hat{o}_t, o_s, o_g), \quad \hat{o}_{t+1} = \mathcal{W}(\hat{o}_t, \hat{a}_{t+1}, o_s, o_g) \tag{1}$$
where $\hat{o}_t$ is the current observation, $\hat{a}_{t+1}$ the action proposed by the planner $\mathcal{P}$, and $\hat{o}_{t+1}$ the next observation visualized by the world model $\mathcal{W}$. The start and goal observations $(o_s, o_g)$ provide the global navigation context. The two modules operate in a closed loop: $\mathcal{P}$ selects $\hat{a}_{t+1}$ conditioned on $\hat{o}_t$ and $(o_s, o_g)$, while $\mathcal{W}$ predicts $\hat{o}_{t+1}$ given $\hat{o}_t$ and $\hat{a}_{t+1}$, which is then fed back into $\mathcal{P}$. This iterative cycle enables imagination-based planning, allowing agents to simulate prospective action–observation trajectories before execution in the real environment. However, the modular training of $\mathcal{P}$ and $\mathcal{W}$ often leads to state–action misalignment, which degrades performance in complex and partially observable settings Ding et al. (2024). (Refer to Appx. A for more related works.)

**Unified World Model with Memory.** To overcome these limitations, we replace the modular pair $(\mathcal{P}, \mathcal{W})$ with a single multimodal backbone, **UniWM**, that couples planning and visualization. A hierarchical memory bank $\mathcal{M}_t$, comprising an intra-step $\mathcal{M}_t^{\mathrm{intra}}$ and a cross-step $\mathcal{M}_t^{\mathrm{cross}}$, fuses short-term evidence with longer-range context (Fig. 2). At each step, UniWM performs:
$$(\hat{a}_{t+1}, \hat{o}_{t+1}) = \mathbf{UniWM}(\hat{o}_t, o_s, o_g, p_0, \mathcal{M}_t), \tag{2}$$
where UniWM simultaneously functions as a **navigation planner** and a **world model**, alternating between these roles to propose actions and visualization outcomes until the goal is reached. Within a unified framework, we instantiate UniWM as a single multimodal large language model (MLLM) $F_\theta$, which interleaves two substeps at each iteration: (i) *action prediction* (planner role) and (ii) *navigation imagination* (world-model role). Both substeps are executed by the same backbone $F_\theta$, jointly trained on planner and world-model data with tailored objectives (Fig. 2(a), Sec. 2.2). During inference, $F_\theta$ is augmented with a hierarchical memory that integrates immediate evidence with longer-horizon context (Fig. 2(b), Sec. 2.3), ensuring temporally consistent predictions across steps.

- **Navigation Planner (Action Prediction):** Given current observation $\hat{o}_t$, conditioned on start and goal observations $(o_s, o_g)$, initial pose $p_0$, and memory bank $\mathcal{M}_t$, $F_\theta$ predicts next action $\hat{a}_{t+1}$:
$$\hat{a}_{t+1} = F_\theta(\hat{o}_t, o_s, o_g, p_0, \mathcal{M}_t). \tag{3}$$

- **World Model (Navigation Visualization):** Given current observation $\hat{o}_t$ and action $\hat{a}_{t+1}$, conditioned on $(o_s, o_g)$, $p_0$, and $\mathcal{M}_t$, $F_\theta$ predicts the next observation $\hat{o}_{t+1}$ after executing $\hat{a}_{t+1}$:
$$\hat{o}_{t+1} = F_\theta(\hat{o}_t, \hat{a}_{t+1}, o_s, o_g, p_0, \mathcal{M}_t). \tag{4}$$

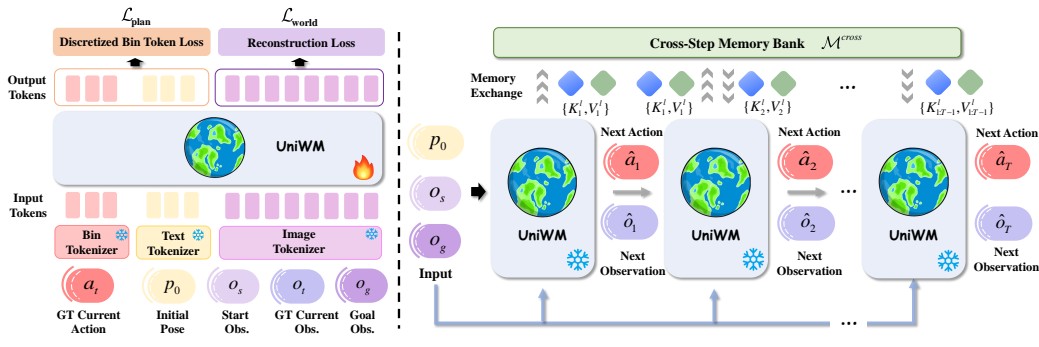

(a) Unified Training          (b) Inference with Memory Bank

Figure 2: **UniWM framework.** **(a) Training:** planner and world-model samples are interleaved within a single unified multimodal autoregressive backbone, optimized jointly with the discretized bin-token loss $\mathcal{L}_{\text{plan}}$ and the reconstruction loss $\mathcal{L}_{\text{world}}$; bin/text/image tokenizers map actions, pose, and observations to tokens. **(b) Inference:** a hierarchical memory supplies intra- and cross-step KV states ($\mathcal{M}_t^{\text{intra}}$ caches the current observation; $\mathcal{M}_t^{\text{cross}}$ accumulates prior steps) to augment attention, yielding robust trajectory-consistent alternating predictions of $\hat{a}_t$ (next action) and $\hat{o}_t$ (next observation). See Fig. 3 for the detailed memory mechanism.

This design allows $F_\theta$ to act jointly as a **navigation planner** and a **world model**, alternating between roles until a terminal Stop is issued. During training, planner and world-model samples are interleaved so that $F_\theta$ learns both behaviors within a single autoregressive framework. At inference, a hierarchical memory bank augments $F_\theta$ by caching key–value states at both intra- and cross-step levels, enabling the integration of immediate observations with longer-range trajectory context. This unified formulation ensures consistent, memory-augmented world modeling throughout navigation.

## 2.2 UNIFIED TRAINING SCHEME

Next, we turn to autoregressive MLLMs that utilize text and image tokens, enabling a unified training scheme for UniWM. We build upon Chameleon and Anole architectures Team (2024); Chern et al. (2024), which integrate a unified Transformer for joint processing of multimodal tokens (Fig. 2 (a)).

**Data Preprocessing.** Each navigation trajectory yields two complementary sample types aligned with Eqs. 3 and 4. For the **navigation planner**, a sample consists of $(o_s, o_g, o_t, p_0)$ with target $\hat{a}_{t+1}$. For the **world model**, inputs additionally include $a_{t+1}$ and the target is $\hat{o}_{t+1}$. Visual observations are encoded as <image> placeholders in structured multimodal prompts, using a sliding window to extract multiple samples per trajectory (See Appx. B.1 for prompt examples). During training, samples from both substeps are interleaved in the same batch to encourage shared representations.

**Multimodal Tokenization.** We employ three tokenizers to unify visual and textual inputs. Following Gafni et al. (2022); Team (2024), a vector-quantized (VQ) image tokenizer discretizes images ($o_s, o_g, o_t$) into visual tokens via a learned codebook, while a byte-pair encoding (BPE) tokenizer Team (2024) encodes pose $p_o$ and text prompts into text tokens. Actions $a_t$ are mapped to discrete bin tokens using the bin tokenizer, which we discuss below. The resulting token sequences are fed to a causal Transformer for joint multimodal modeling.

**Training Objective.** To optimize our model for the distinct characteristics of the navigation planner and world model, we introduce tailored training objectives. At each iteration, our autoregressive MLLM jointly processes samples from both roles, producing logits across the unified vocabulary.

• **Discretized Bin Token Loss (Navigation Planner).** We propose a new classification-based approach for training the planner, which formulates continuous action prediction as multi-class classification over discretized motion bins. Each navigation action $a_t \in \mathbb{R}^3$ is represented as $(x_t, y_t, \phi_t)$, where $x_t$ and $y_t$ denote planar translations and $\phi_t$ denotes yaw rotation. We uniformly partition each dimension into fixed-size bins with size $b = 0.01$, computing bin index as $\lfloor |v|/b \rfloor$ for value $v$. We use separate positive and negative token prefixes to encode the sign and another prefix for the target dimension. For example, $x$-axis translation with $v = 0.03$ is encoded as <dx_pos_bin_03>. This scheme represents all three dimensions as special bin tokens from disjoint token sets: $\mathcal{T}_x, \mathcal{T}_y$, and $\mathcal{T}_\phi$. Let $P(t_i)$ denote the model's predicted distribution over all vocabulary tokens at decoding position $i$. We supervise the planner using **discretized bin token loss** over each action dimension:

$$\mathcal{L}_{\text{plan}} = \frac{1}{3} \sum_{k \in \{x,y,\phi\}} \left( -\log P\big(t_i = t_k^* \,\big|\, t_i \in \mathcal{T}_k\big) \right) + \mathcal{L}_{\text{CE}}, \tag{5}$$

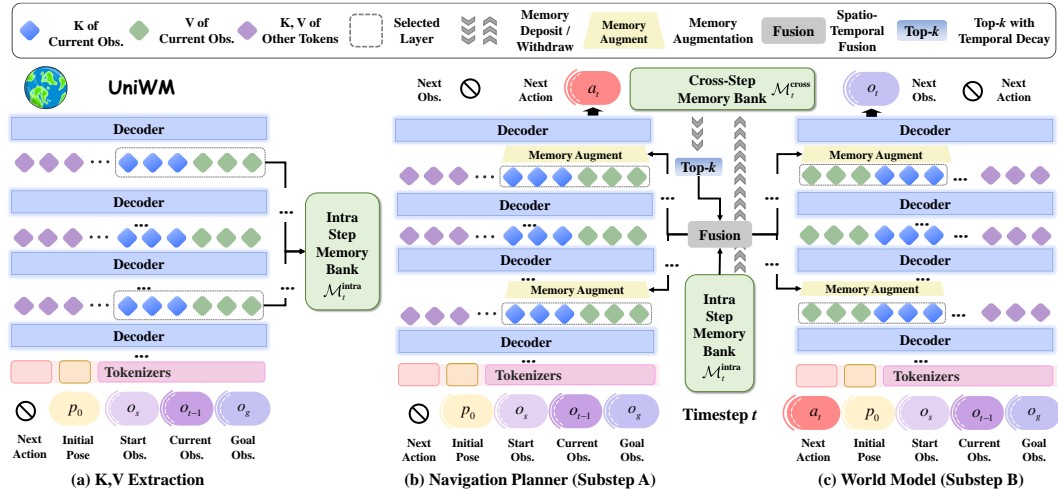

Figure 3: **Overview of hierarchical memory bank mechanism** ($\mathcal{M}_t^{\text{intra}}$ & $\mathcal{M}_t^{\text{cross}}$). **(a)** $KV$ (keys/values) extracted from selected layers are deposited into $\mathcal{M}_t^{\text{intra}}$ at the beginning of each step $t$ (Eq. 7). **(b)(c)** $\mathcal{M}_t^{\text{intra}}$ is merged with the accumulated cross-step memory $\mathcal{M}_t^{\text{cross}}$ via top-$k$ similarity gating (Eq. 8) and exponential temporal decay (Eq. 9), yielding a fused memory (Eq. 10) that augments attention for both the planner and the world-model substeps (Eq. 11) to promote trajectory-consistent predictions. At the end of step $t$, $\mathcal{M}_t^{\text{intra}}$ (with timestamp $t$) is appended to $\mathcal{M}_t^{\text{cross}}$ for reliable reuse at step $t+1$, enabling robustly efficient rollouts.

where $t_k^*$ is the ground-truth bin token in dimension $k$, and $\mathcal{L}_{\text{CE}}$ is the cross-entropy loss for output text tokens as output may also include text action $\texttt{Stop}$.

• **Reconstruction Loss (World Model).** We introduce a **reconstruction loss** to enforce fidelity in the predicted future observations to encourage accurate navigation visualization. Given ground-truth visual embedding $\mathbf{v}_i$ for token $i$ (out of $n$ tokens in the next observation $\hat{o}_{t+1}$) and the visual codebook embeddings $\mathcal{E} = \{\mathbf{v}_1, \ldots, \mathbf{v}_N\}$ where $N$ is the total number of visual token vocabulary:

$$\mathcal{L}_{\text{world}} = \frac{1}{n} \sum_{i=1}^{n} \|\mathbf{v}_i, \mathcal{E}\|^2 \cdot P(t_i), \qquad (6)$$

where $\sum_{i=1}^{n} \|\mathbf{v}_i, \mathcal{E}\|^2$ is the similarity vector indicating distances between $\mathbf{v}_i$ and all codebook embeddings, with lower similarity referring to larger distances, and $P(t_i) \in \mathbb{R}^{1 \times N}$ denotes predicted probability distribution over visual tokens at position $i$. Throughout training, all tokenizers remain frozen, and only Transformer parameters are updated under autoregressive next-token prediction.

## 2.3 Inference with Memory Bank

At the inference phase, UniWM alternates between two substeps: action prediction and navigation visualization. As illustrated in Fig. 3 and Alg. 1, UniWM employs a hierarchical two-level memory bank mechanism. The **intra-step** memory $\mathcal{M}_t^{\text{intra}}$ caches key, value $(K, V)$ pairs extracted from the current observation $\hat{o}_{t-1}$ at selected Transformer decoder layers, while the **cross-step** memory $\mathcal{M}_t^{\text{cross}}$ accumulates all past intra-step memories $\mathcal{M}_m^{\text{intra}}$, where $(m \in 1, \ldots, t-1)$ together with their associated step indices $t_m$. This design allows $\mathcal{M}_t^{\text{cross}}$ to maintain a persistent trajectory-level context, thereby enabling $F_\theta$ to integrate both short-term and longer-term dependencies across steps.

**Two-level Cache Design.** At the beginning of each step $t$, the intra-step memory bank $\mathcal{M}_t^{\text{intra}}$ is reset to avoid contamination from the previous step: $\mathcal{M}_t^{\text{intra}} \leftarrow \varnothing$. Given the tokenized multimodal input, we identity the span of the current observation $\hat{o}_{t-1}$ by marking its token sequence with two special boundary tokens, $\texttt{<boss>}$ and $\texttt{<eoss>}$, thereby yielding the index set $\mathcal{I}_t$. We then extract $K, V$ pairs to form $\mathcal{M}_t^{\text{intra}}$ only from this span at a selected subset of decoder layers $L_{\text{save}} = \{l_0, \ldots, l_{31}\}$:

$$\mathcal{M}_t^{\text{intra}} = \{K_t^{(l)}, V_t^{(l)}\} = \{f_K^{(l)}(\mathbf{x}_{\mathcal{I}_t}), f_V^{(l)}(\mathbf{x}_{\mathcal{I}_t})\}, \quad where\ l \in L_{\text{save}}, \qquad (7)$$

where $\{K_t^{(l)}, V_t^{(l)}\}$ denotes keys and values obtained from the $l$-th decoder layer at step $t$, $\mathbf{x}$ represents the hidden states of the multimodal input sequence at that layer, $\mathbf{x}_{\mathcal{I}_t}$ refers to the slice of hidden states indexed by $\mathcal{I}_t$, and $f_K^{(l)}$ and $f_V^{(l)}$ are the key and value projection mappings in layer $l$. In parallel, as demonstrated in Fig. 3, the cross-step memory $\mathcal{M}_t^{\text{cross}}$ aggregates selected intra-step caches from previous $t-1$ steps with timestamps $t_m$: $\mathcal{M}_t^{\text{cross}} = \{(K_m^{(l)}, V_m^{(l)}, t_m)\}_{l \in L_{\text{save}}}$.

**Spatio-temporal Fusion.** At each action prediction substep of step $t$, the intra-step memory $\mathcal{M}_t^{\text{intra}}$ is merged with the accumulated cross-step memory $\mathcal{M}_t^{\text{cross}}$ to construct a fused memory $\tilde{\mathcal{M}}_t$, which subsequently enhances the attention mechanism for both substeps. This fusion incorporates spatial similarity selection and temporal recency weighting as shown in Fig. 3:

*(i) Similarity gating.* We flatten both current and historical keys and compute entry-wise cosine similarity $s_m^{(l)}$. The indices of the top-$k$ most similar entries are collected into the set $h_t^{(l)}$:

$$s_m^{(l)} = \cos\left(K_t^{(l)}, K_m^{(l)}\right), \qquad h_t^{(l)} = \text{top-}k\left(s_m^{(l)}\right), \quad where\ m \in \{1, \dots, t-1\}. \tag{8}$$

*(ii) Temporal decay.* Each selected entry is weighted by an exponential decay factor determined by its recency gap $\Delta t_m = t - t_m$, that larger weights correspond to stronger influence on subsequent predictions. Here we set $\gamma = 0.2$, which biases the weighting toward more recent steps:

$$\alpha_m^{(l)} = \frac{\exp\left(-\gamma\,\Delta t_m\right)}{\sum_{j \in h_t^{(l)}} \exp\left(-\gamma\,\Delta t_j\right)}. \tag{9}$$

*(iii) Memory fusion.* The fused memory $\tilde{\mathcal{M}}_t = \{\tilde{K}_t^{(l)}, \tilde{V}_t^{(l)}\}_{l \in L_{\text{save}}}$ is formed by concatenating the current intra-step memory with the weighted historical entries so that historical contributions are explicitly modulated by both spatial similarity and temporal recency:

$$\tilde{K}_t^{(l)} = \textbf{Concat}\left(K_t^{(l)}, \alpha_h^{(l)} K_h^{(l)}\right), \tilde{V}_t^{(l)} = \textbf{Concat}\left(V_t^{(l)}, \alpha_h^{(l)} V_h^{(l)}\right), where\ h \in h_t^{(l)},\ l \in L_{\text{save}}. \tag{10}$$

**Memory-augmented Attention.** The fused memory $\tilde{\mathcal{M}}_t$ then directly engage in cross-attention computation. The attention mechanism can be formally described as scaled dot-product attention:

$$\tilde{Q}_t^{(l)} = \text{Att}(Q_t^{(l)}, \tilde{K}_t^{(l)}, \tilde{V}_t^{(l)}) = \text{softmax}\left(\frac{Q_t^{(l)} \tilde{K}_t^{(l)\top}}{\sqrt{d_k}}\right) \tilde{V}_t^{(l)}, \tag{11}$$

where $Q_t^{(l)}$ denotes the current query at layer $l$, and $d_k$ is the key dimension. $\tilde{Q}_t^{(l)}$ subsequently propagate through subsequent predictions. This mechanism equips UniWM with trajectory-consistent reasoning by leveraging both current observations and temporally structured historical memories.

**Rollout Procedure.** The full inference process of one trajectory can be summarized as: at each step $t$, UniWM resets the intra-step memory $\mathcal{M}_t^{\text{intra}}$, extracts $K$, $V$ pairs from the current observation $\hat{o}_{t-1}$ (Eq. 7), and fuses them with the cross-step memory $\mathcal{M}_t^{\text{cross}}$ using similarity gating (Eq. 8) and temporal decay (Eq. 9) to form $\tilde{\mathcal{M}}_t$ (Eq. 10). The planner then predicts the next action $\hat{a}_t$ using enhanced attentions (Eqs. 3 and 11), and the world model generates the next observation $\hat{o}_t$ (Eqs. 4 and 11). Finally, $\mathcal{M}_t^{\text{intra}}$ with its timestamp $t$ is appended to $\mathcal{M}_t^{\text{cross}}$ for future use. This process iterates until a Stop action is emitted to terminate this navigation trajectory.

## 3 EXPERIMENTS

**Datasets.** We use four robotics datasets (**Go Stanford** Hirose et al. (2018), **ReCon** Shah et al. (2021), **SCAND** Karnan et al. (2022), and **HuRoN** Hirose et al. (2023)) for training and in-domain evaluation, and reserve **TartanDrive** Triest et al. (2022) as an unseen test set. We select these datasets because they cover complementary aspects of real-world navigation; for instance, ReCon targets open-world settings, whereas SCAND emphasizes socially compliant navigation across varied environments. We choose TartanDrive for unseen evaluation because it uniquely contains visible robot body structures and embodiment-related artifacts that evolve over time, which better reflects real-world distribution shift. To standardize action magnitudes across embodiments, we normalize per-frame displacement by average step size (in meters), filter out backward motions like Bar et al. (2025); Sridhar et al. (2024) and trajectories shorter than three steps, and segment each trajectory's visual stream into semantically coherent sub-scenes using Qwen-VL-2.5 Bai et al. (2025). After filtering, we obtain the following numbers of trajectories: Go Stanford (train/eval: 4457/496), ReCon (4652/517), SCAND (2560/285), HuRoN (4642/516), and TartanDrive (eval only: 500).

**Evaluation Metrics.** We evaluate performance using two suites of metrics. **(1) Navigation quality.** We report Absolute Trajectory Error (ATE), Relative Pose Error (RPE) Sturm et al. (2012), and Success Rate (SR). SR deems a trajectory successful if final distance to goal is smaller than the agent's average step size (in meters). **(2) Visualization quality.** For navigation visualization, we use structural/perceptual metrics SSIM Wang et al. (2004), PSNR Hore & Ziou (2010), LPIPS Zhang et al. (2018), and DreamSim Fu et al. (2023). To assess longer-horizon stability under rollout, we introduce four metrics: SSIM@n, PSNR@n, LPIPS@n, and DreamSim@n. (Details in Appx. C.1.)

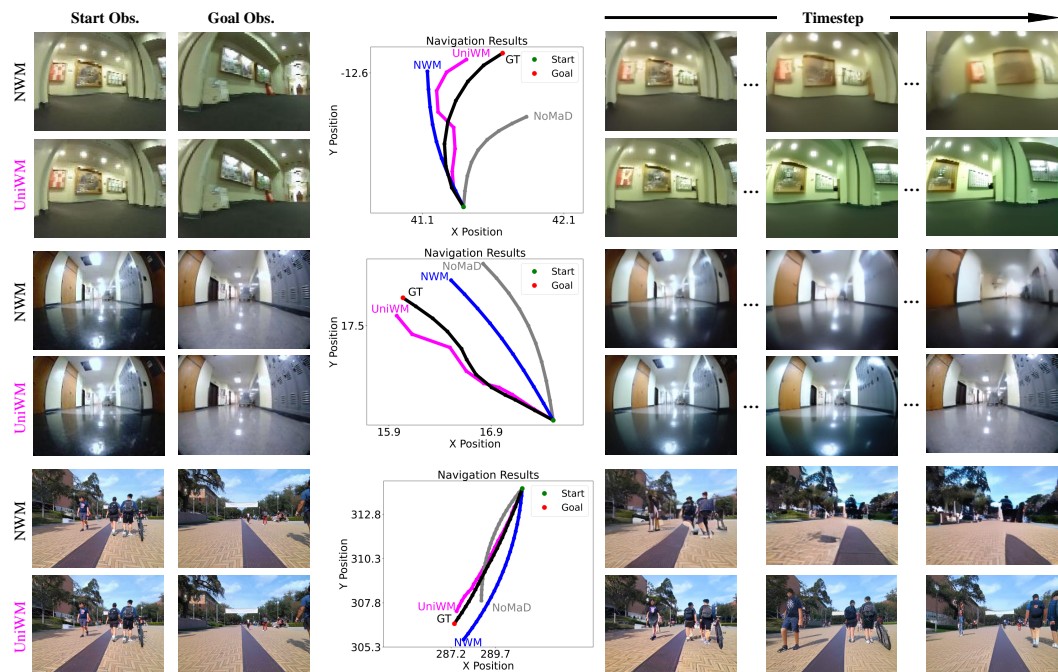

Figure 4: **Qualitative Comparisons** on Go Stanford and HuRoN datasets across UniWM, NWM, and No-MaD. Qualitative results here include both static indoor environments and outdoor scenarios with moving pedestrians. The central trajectory plots highlight difference between predicted $A_T$ and the ground-truth.

Table 1: **Comparison with SOTA Methods upon Goal-Conditioned Visual Navigation** on evaluation splits of Go Stanford, ReCon, SCAND, and HuRoN with SR, ATE, and RPE.

| Method | Go Stanford | | | ReCon | | | SCAND | | | HuRoN | | |
|---|---|---|---|---|---|---|---|---|---|---|---|---|
| | SR ↑ | ATE ↓ | RPE ↓ | SR ↑ | ATE ↓ | RPE ↓ | SR ↑ | ATE ↓ | RPE ↓ | SR ↑ | ATE ↓ | RPE ↓ |
| GNM Shah et al. (2022) | 0.27 | 1.11 | 0.31 | 0.72 | 0.70 | 0.20 | 0.49 | 0.51 | 0.21 | 0.36 | 1.07 | 0.35 |
| VINT Shah et al. (2023) | 0.29 | 1.09 | 0.35 | 0.68 | 0.84 | 0.28 | 0.45 | 0.58 | 0.28 | 0.30 | 1.19 | 0.43 |
| NoMaD Sridhar et al. (2024) | 0.33 | 0.94 | 0.30 | 0.71 | 0.77 | 0.21 | 0.50 | 0.54 | 0.23 | 0.37 | 0.92 | 0.33 |
| Anole-7B Chern et al. (2024) | 0.18 | 2.18 | 0.73 | 0.41 | 1.74 | 0.69 | 0.29 | 1.37 | 0.71 | 0.20 | 1.92 | 0.78 |
| NWM Bar et al. (2025) | 0.45 | 0.80 | 0.27 | 0.79 | 0.58 | 0.17 | 0.55 | 0.41 | 0.19 | 0.41 | 0.73 | 0.28 |
| **UniWM (w/o $\mathcal{M}$)** | 0.71 | 0.32 | 0.10 | 0.82 | 0.35 | 0.12 | 0.61 | 0.36 | 0.14 | 0.70 | 0.42 | 0.15 |
| **UniWM (with only $\mathcal{M}_t^{intra}$)** | 0.73 | 0.29 | **0.09** | 0.85 | 0.38 | 0.13 | 0.64 | 0.33 | **0.13** | 0.74 | 0.44 | 0.15 |
| **UniWM (with $\mathcal{M}_t^{intra}$ & $\mathcal{M}_t^{cross}$)** | **0.75** | **0.22** | **0.09** | **0.93** | **0.34** | **0.11** | **0.68** | **0.32** | **0.13** | **0.76** | **0.38** | **0.13** |

**Implementation Details.** UniWM is fine-tuned upon GAIR Anole-7B Chern et al. (2024) (4096-token context) while freezing the text and image tokenizers as well as the bin-token encoder. Input images are resized to $448 \times 448$ (height × width) and discretized into 784 visual tokens. During training, only the LoRA Hu et al. (2022) adapters (rank = 16) in the Transformer's $qkv$-projections are updated Liu et al. (2023). Optimization is performed with AdamW for 20 epochs using a learning rate of $2 \times 10^{-4}$. Training runs on $4\times$NVIDIA A100 GPUs (80GB each) with a global batch size of 8 (per-GPU batch = 1, gradient accumulation = 2). For inference, we designate two special tokens <boss> and <eoss> (token IDs 8196 and 8197) to trigger the key–value (KV) deposit of intra-step memory bank, and extract KV for enhancement from decoder layers $\{l_0, l_7, l_{15}, l_{23}, l_{31}\}$. We retrain all diffusion-based baselines, including NWM, from scratch using the same four training datasets as UniWM (Go Stanford, ReCon, SCAND, HuRoN), and evaluated zero-shot on TartanDrive. Thus, UniWM and NWM (as well as other baselines) are trained under identical conditions.

## 3.1 COMPARISON TO STATE-OF-THE-ART METHODS

**Navigation performance.** Table 1 reports goal-conditioned visual navigation results on four in-domain datasets (Go Stanford, ReCon, SCAND, and HuRoN), while Fig. 4 provides qualitative comparison results (more results in Appx. C.3). We compare UniWM with traditional navigation policies GNM Shah et al. (2022), VINT Shah et al. (2023), NoMaD Sridhar et al. (2024), and Anole-7B Chern et al. (2024) under direct prompting (zero-shot). We include NWM Bar et al. (2025), which leverages world modeling through CDiT within MPC framework. UniWM consistently delivers superior results compared to all SOTA baselines. Without memory augmentation, UniWM

Table 2: **Comparison with SOTA methods on visualization performance**, averaged over evaluation splits of Go Stanford, ReCon, SCAND, and HuRoN.

| Method | SSIM ↑ | PSNR ↑ | LPIPS ↓ | DreamSIM ↓ | SSIM@5 ↑ | PSNR@5 ↑ | LPIPS@5 ↓ | DreamSIM@5 ↓ |
|---|---|---|---|---|---|---|---|---|
| Diamond Alonso et al. (2024) | 0.311 | 9.837 | 0.410 | 0.131 | 0.186 | 6.352 | 0.582 | 0.252 |
| NWM Bar et al. (2025) | 0.389 | 11.420 | 0.318 | 0.089 | 0.256 | 7.755 | 0.494 | 0.174 |
| **UniWM** | **0.457** | **13.607** | **0.254** | **0.041** | **0.350** | **10.874** | **0.435** | **0.126** |

Table 3: **Impact of Context Size and Image Token Length** on both navigation and visualization performance, averaged over four datasets. All settings are evaluated without memory banks.

| Context | Token Len. | Navigation | | | Visualization | | | | | |
|---|---|---|---|---|---|---|---|---|---|---|
| | | SR ↑ | ATE ↓ | RPE ↓ | SSIM ↑ | LPIPS ↓ | DreamSIM ↓ | SSIM@5 ↑ | LPIPS@5 ↓ | DreamSIM@5 ↓ |
| 1 | 784 | **0.71** | **0.36** | **0.13** | **0.457** | 0.254 | **0.041** | **0.350** | **0.435** | **0.126** |
| 2 | 625 | 0.68 | 0.39 | 0.15 | 0.448 | **0.247** | 0.051 | 0.336 | 0.451 | 0.137 |
| 2 | 484 | 0.55 | 0.53 | 0.26 | 0.365 | 0.328 | 0.084 | 0.258 | 0.515 | 0.192 |
| 4 | 484 | 0.64 | 0.44 | 0.19 | 0.425 | 0.285 | 0.052 | 0.315 | 0.462 | 0.141 |

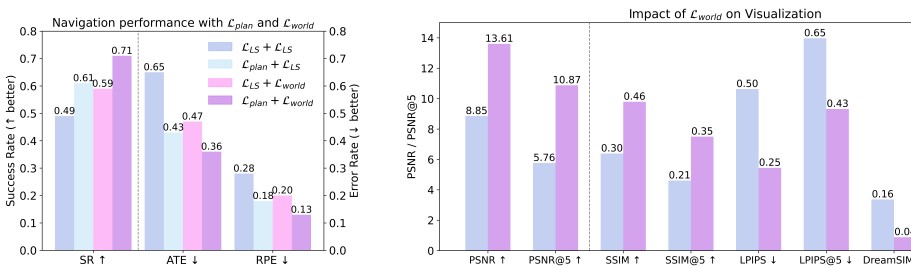

Figure 5: **Impact of discretized bin-token loss ($\mathcal{L}_{\text{plan}}$) and reconstruction Loss ($\mathcal{L}_{\text{world}}$)** on navigation (left) and visualization (right) performance, averaged over evaluation splits of Go Stanford, ReCon, SCAND, and HuRoN. X-axis arrows indicate whether higher or lower values are preferable.

achieves substantial gains in SR (e.g., 0.71 vs. 0.45 for NWM on Go Stanford) and ATE/RPE across datasets. Equipping UniWM with intra-step memory stabilizes predictions, while cross-step memory enhances longer-horizon consistency, leading to best overall performance.

**Visualization performance.** Table 2 evaluates UniWM's visualization ability. We compare against Diamond Alonso et al. (2024), a diffusion-based world model on UNet, and NWM Bar et al. (2025) with CDiT. UniWM achieves competitive results across all metrics. On one-step predictions, it delivers highest structural similarity (SSIM = 0.457) and perceptual alignment (DreamSim = 0.041). Under open-loop rollouts, UniWM maintains stability with SSIM@5 = 0.350, preserving semantic consistency and mitigating compounding errors in longer-horizon evaluations.

## 3.2 Ablation Studies

### 1. How do context size and token length affect navigation and visualization performance?

Table 3 analyzes varying context size and token length per image. Since Anole-7B has a fixed 4096 token context window, increasing context frames requires reducing tokens per frame, creating a trade-off between temporal coverage and spatial resolution. Both navigation and visualization performance improve as context size or token length increases (e.g., (2×484,4×484) and (2×484, 2×625). Comparing 1×784 with 2×625 and 4×484 shows higher token length can outweigh additional context, suggesting spatial resolution has stronger overall impact under fixed token budget.

### 2. Do discretized bin token loss $\mathcal{L}_{\text{plan}}$ and reconstruction loss $\mathcal{L}_{\text{world}}$ help training?

To evaluate the impact of different training objectives on navigation and visualization performance, we compare $\mathcal{L}_{\text{plan}}$ for action tokens and $\mathcal{L}_{\text{world}}$ for image tokens against a label smoothing loss ($\mathcal{L}_{\text{LS}}$). Note that $\mathcal{L}_{\text{LS}}$ is used only in this ablation study as a baseline, not in our final training recipe. From Fig. 5 (right), replacing $\mathcal{L}_{\text{LS}}$ with $\mathcal{L}_{\text{world}}$ on image tokens markedly enhances visualization quality and gains persist under rollout, while Fig. 5 (left) demonstrates that both $\mathcal{L}_{\text{world}}$ and $\mathcal{L}_{\text{plan}}$ benefit navigation. Fig. 5 (left) also shows the best performance arises when combining $\mathcal{L}_{\text{plan}}$ with $\mathcal{L}_{\text{world}}$, and the navigation gains from $\mathcal{L}_{\text{plan}}$ are larger than those from $\mathcal{L}_{\text{world}}$ under matched conditions (SR +0.12 vs. +0.10). A plausible explanation is that $\mathcal{L}_{\text{world}}$ improves navigation indirectly by enhancing visualization quality, whereas $\mathcal{L}_{\text{plan}}$ optimizes UniWM's action decisions directly.

### 3. Should the navigation planner and world model be trained jointly?

Table 4: **Comparison of the unified UniWM and the separate planner + world-model setting** on both navigation (SR, ATE, RPE) and visualization metrics (SSIM, LPIPS, DreamSim).

| Method | SR ↑ | ATE ↓ | RPE ↓ | SSIM ↑ | LPIPS ↓ | DreamSIM ↓ | SSIM@5 ↑ | LPIPS@5 ↓ | DreamSIM@5 ↓ |
|---|---|---|---|---|---|---|---|---|---|
| UniWM (Separate) | 0.65 | 0.41 | 0.16 | 0.443 | 0.280 | 0.055 | 0.329 | 0.470 | 0.154 |
| UniWM | **0.71** | **0.36** | **0.13** | **0.457** | **0.254** | **0.041** | **0.350** | **0.435** | **0.126** |

Table 5: **Impact of number of selected layers** included in memory bank on navigation performance of UniWM (with $\mathcal{M}_t^{\text{intra}}$ & $\mathcal{M}_t^{\text{cross}}$) on evaluation splits of four in-domain datasets.

| Layer Num | Go Stanford | | | ReCon | | | SCAND | | | HuRoN | | |
|---|---|---|---|---|---|---|---|---|---|---|---|---|
| | SR ↑ | ATE ↓ | RPE ↓ | SR ↑ | ATE ↓ | RPE ↓ | SR ↑ | ATE ↓ | RPE ↓ | SR ↑ | ATE ↓ | RPE ↓ |
| 1 | 0.71 | 0.30 | 0.10 | 0.84 | 0.36 | 0.12 | 0.62 | 0.35 | 0.14 | 0.72 | 0.41 | 0.14 |
| 3 | 0.74 | 0.27 | 0.09 | 0.89 | 0.35 | 0.11 | 0.66 | 0.33 | 0.13 | 0.75 | 0.39 | 0.14 |
| 5 | **0.75** | **0.22** | **0.09** | **0.93** | **0.34** | **0.11** | 0.68 | 0.32 | **0.13** | **0.76** | **0.38** | **0.13** |
| 7 | 0.74 | 0.25 | 0.09 | 0.91 | 0.35 | 0.12 | **0.69** | **0.31** | 0.13 | 0.74 | 0.38 | 0.14 |
| 16 | 0.61 | 0.46 | 0.23 | 0.70 | 0.58 | 0.26 | 0.52 | 0.49 | 0.24 | 0.57 | 0.55 | 0.22 |
| 32 | 0.58 | 0.52 | 0.26 | 0.67 | 0.64 | 0.29 | 0.49 | 0.55 | 0.27 | 0.54 | 0.61 | 0.25 |

Table 6: **Comparison of navigation performance under different step strategies** across four datasets.

| Step Strategy | Go Stanford | | | ReCon | | | SCAND | | | HuRoN | | |
|---|---|---|---|---|---|---|---|---|---|---|---|---|
| | SR ↑ | ATE ↓ | RPE ↓ | SR ↑ | ATE ↓ | RPE ↓ | SR ↑ | ATE ↓ | RPE ↓ | SR ↑ | ATE ↓ | RPE ↓ |
| Predict both | 0.65 | 0.38 | 0.13 | 0.80 | 0.41 | 0.15 | 0.57 | 0.39 | 0.17 | 0.63 | 0.47 | 0.20 |
| Interleave | **0.71** | **0.32** | **0.10** | **0.82** | **0.35** | **0.12** | **0.61** | **0.36** | **0.14** | **0.70** | **0.42** | **0.15** |

We conduct an ablation in which the planner and the world model are trained separately using the same data and schedule, and compare them with our unified model in Table 4. The unified UniWM consistently outperforms the separate planner–world model setup across both navigation and visualization metrics, providing direct empirical evidence that the joint architecture and training more effectively align imagination with control.

**4. Do we need both intra-step and cross-step memory bank during inference?**

We compare using three UniWM variants in Table 1: no memory, intra-step memory bank only, and intra+cross memory banks. Adding intra-step memory improves SR on all four datasets and generally stabilizes pose estimates—RPE decreases or remains comparable. Further augmenting with cross-step memory yields the best SR and RPE (overall: 0.78 / 0.11) across all datasets. These results indicate that both intra-step and cross-step memories are important at inference time, with cross-step memory providing longer-horizon gains on top of intra-step stabilization.

**5. How does the number of selected layers included in the memory bank affect inference?**

We evaluate the impact of integrating memory at different numbers of selected layers (with both $\mathcal{M}_t^{\text{intra}}$ and $\mathcal{M}_t^{\text{cross}}$ enabled) and report navigation metrics in Table 5. Increasing from a single shallow layer to a moderate multi-depth integration (3–7 layers) progressively improves SR/ATE/RPE, with the 5-layer setting achieving strong results across all four datasets, indicating that multi-depth integration facilitates iterative feature refinement through the network hierarchy. In contrast, dense integration (16–32 layers) degrades performance and incurs higher compute and KV overhead. Balancing accuracy and efficiency, we adopt the 5-layer configuration for inference.

**6. Does goal conditioning affect generalization?**

Table 7 explores whether conditioning on the goal image compromise generalization in unseen environments. We retrain UniWM from scratch on the same four datasets, but remove goal image from the navigation visualization substep during both training and inference, keeping all other settings unchanged. As shown in Table 7 (we evaluate

Table 7: **Ablation of goal conditioning** with evaluated on TartanDrive (unseen).

| Method | SR ↑ | ATE ↓ | RPE ↓ |
|---|---|---|---|
| UniWM (w/o $o_g$ (retrained)) | 0.33 | 1.37 | 0.51 |
| UniWM | **0.35** | **1.20** | **0.46** |

on unseen TartanDrive split), original UniWM still outperforms the retrained variant in unseen environments, indicating that conditioning on the goal image does not harm generalization in our task.

**7. Why UniWM predict action and observation at different substep?**

Table 6 compares two step strategies of UniWM: predict both vs. interleave (our choice). In predict both, each training sample contains $(o_s, o_g, o_t, p_0)$ and the model jointly predicts the next action $\hat{a}_{t+1}$ and next observation $\hat{o}_{t+1}$ in a single forward pass. In interleave, we provide two sample types corresponding to the planner and world-model sub-steps and alternate them; inference follows

Table 8: **Zero-shot navigation performance** evaluated on TartanDrive (unseen).

| Method | SR ↑ | ATE ↓ | RPE ↓ |
|---|---|---|---|
| GNM Shah et al. (2022) | 0.16 | 2.45 | 0.79 |
| VINT Shah et al. (2023) | 0.13 | 2.38 | 0.79 |
| NoMaD Sridhar et al. (2024) | 0.18 | 2.23 | 0.77 |
| Anole-7B Chern et al. (2024) | 0.15 | 2.12 | 0.83 |
| NWM Bar et al. (2025) | 0.27 | 1.61 | 0.62 |
| **UniWM (w/o $\mathcal{M}$)** | 0.35 | 1.20 | 0.46 |
| **UniWM (with only $\mathcal{M}_t^{\text{intra}}$)** | 0.38 | 1.04 | 0.41 |
| **UniWM (with $\mathcal{M}_t^{\text{intra}}$ & $\mathcal{M}_t^{\text{cross}}$)** | **0.42** | **0.95** | **0.37** |

Figure 6: **Qualitative Results** in unseen environments (Tartan-Drive) with UniWM. Red boxes denote ego-robot parts.

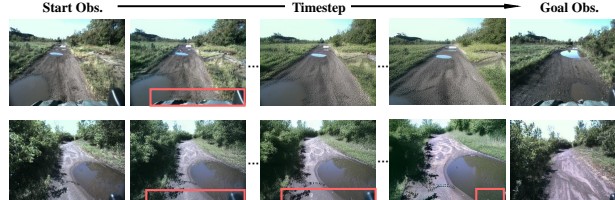

the same alternation. Across all datasets, interleave yields higher SR and lower ATE/RPE, which empirically verifies UniWM's design choice to predict actions and observations in different substeps.

## 3.3 GENERALIZATION IN UNSEEN ENVIRONMENTS

We evaluate zero-shot generalization on the unseen TartanDrive split without any fine-tuning in Table 8 and Fig 6. UniWM consistently delivers competitive results compared to all baselines. Even without memory augmentation, UniWM achieves substantial gains in SR and reduces pose errors. Equipping UniWM with $\mathcal{M}_t^{\text{intra}}$ further stabilizes predictions, while adding $\mathcal{M}_t^{\text{cross}}$ enhances longer-horizon consistency, yielding the best overall performance (SR 0.42, ATE 0.95, RPE 0.37). These results confirm strong generalization of UniWM in unseen environments.

**Error cases and limitations analysis.** On TartanDrive, egocentric observation occasionally contains visible ego-robot parts (e.g., bumper/hood). Fig. 6 shows UniWM's first-step prediction preserves these ego cues, but during rollouts they fade and disappear. We attribute this to domain gap: our training sets lack visible ego-robot regions, so the model treats them as background and "inpaints" them away. This causes inconsistencies with ground-truth frames in unseen environments.

## 4 CONCLUSION

We present UniWM, a unified memory-augmented world model that couples visual imagination with navigation planning in a single multimodal autoregressive architecture. By jointly modeling perception, prediction, and planning, UniWM closes state–action misalignment. A hierarchical memory fuses short-term observations with longer-range context, stabilizing longer-horizon rollouts. Experiments across four benchmarks and zero-shot evaluation on the TartanDrive dataset show higher SR and lower ATE/RPE than strong baselines. UniWM represents a promising direction toward scalable and generalizable visual navigation systems. Current limitations include domain shift (e.g., ego-robot artifacts) and a fixed token budget, which future work can address through adaptive token allocation, uncertainty-aware planning, and closed-loop deployment on real robots.

## ETHICS STATEMENT

This work adheres to the ICLR Code of Ethics. In this study, no human subjects or animal experimentation was involved. All datasets used, including Go Stanford, ReCon, SCAND, HuRoN and TartanDrive, were sourced in compliance with relevant usage guidelines, ensuring no violation of privacy. We have taken care to avoid any biases or discriminatory outcomes in our research process. No personally identifiable information was used, and no experiments were conducted that could raise privacy or security concerns. We are committed to maintaining transparency and integrity throughout the research process.

## REPRODUCIBILITY STATEMENT

We have made every effort to ensure that the results presented in this paper are reproducible. The experimental setup, including training steps, model configurations, and hardware details, is described in detail in the paper. We have also provided a full description of training and evaluation setup in supplementary, to assist others in reproducing our experiments.

Additionally, Go Stanford, ReCon, SCAND, HuRoN and TartanDrive datasets are publicly available, ensuring consistent and reproducible evaluation results. We believe these measures will enable other researchers to reproduce our work and further advance the field.

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

# APPENDIX

## A  RELATED WORK

World models have emerged as a unifying paradigm for learning predictive representations of environment dynamics, with applications spanning simulation, decision-making, and embodied navigation. In this section, we review two lines of related research: (i) advances in generic world model architectures, and (ii) their application to goal-conditioned visual navigation.

**World Models.** World models Ha & Schmidhuber (2018) have become a central paradigm for learning predictive representations of environment dynamics, evolving from compact recurrent structures to large-scale generative and multimodal systems. Early works such as World Model Ha & Schmidhuber (2018) and Dreamer Hafner et al. (2019; 2022; 2024) employed RNN-based latent dynamics to capture temporal transitions. Transformer-based designs (I-JEPA Assran et al.

(2023), V-JEPA Bardes et al. (2024), DINO-WM Baldassarre et al. (2025)) introduced scalable attention mechanisms for richer spatio-temporal abstraction. More recently, diffusion-based generators (Sora Brooks et al. (2024), Cosmos Agarwal et al. (2025), Genie Bruce et al. (2024)) have been adapted for environment dynamics, enabling high-fidelity simulation and downstream planning Alonso et al. (2024); Valevski et al. (2024); Bar et al. (2025); Yu et al. (2025), though at the cost of efficiency and limited integration with policy learning Xiao et al. (2025). Parallel efforts exploit LLMs to simulate dynamics via prompting Zhao et al. (2025); Xing et al. (2025), but face modality misalignment, temporal inconsistency, and grounding challenges, with context length limits causing memory degradation over longer horizons. Our work builds on these advances by introducing structured memory banks and a unified paradigm that jointly couples perception, prediction, and decision-making, mitigating the alignment and stability issues in prior modular designs.

**World Models for Navigation.** Goal-conditioned navigation is a natural testbed for world models, as it requires tight coupling between perception and policy Frey et al. (2023). Policy-centric methods Shah et al. (2022; 2023); Sridhar et al. (2024) map observations directly to actions without explicitly modeling environment dynamics. In contrast, navigation-oriented world models predict future observations to support temporally informed planning Yao et al. (2025). Early works such as PathDreamer Koh et al. (2021) used GANs to simulate indoor vision–language navigation but depended on auxiliary inputs (e.g., semantic maps), limiting generalization Lin et al. (2024). More recent approaches (e.g., NWM Bar et al. (2025)) integrate raw video prediction into the navigation loop to produce realistic rollouts, yet still decouple planning from perception, relying on separate policy modules and failing to reason jointly over actions and observations. Building on these advances, we propose a unified multimodal backbone that aligns action prediction with observation imagination, enabling end-to-end navigation through temporally grounded dynamics modeling.

## B  METHOD DETAILS

### B.1  PROMPT DESIGN AND EXAMPLES

We examine the detailed prompt formulation and response behaviors of two substeps: action prediction and navigation visualization in Figs. 7 and 8. These examples illustrate how multimodal inputs guide both the navigation planner and the world model in visually grounded navigation.

### B.2  PSEUDO-CODE FOR HIERARCHICAL MEMORY BANK MECHANISM

Alg. 1 details the inference process of UniWM, which systematically employs the hierarchical memory bank. The algorithm begins by initializing the intra-step memory $\mathcal{M}_t^{\text{intra}}$ and the persistent cross-step memory $\mathcal{M}_t^{\text{cross}}$ as empty sets (Line 9). It also defines a subset of decoder layers, $L_{\text{save}}$, from which Key-Value (KV) pairs will be extracted (Line 10). The main logic operates in a loop for each step $t$ from 1 to $T$ (Line 12), divided into two substeps:

**Action Prediction.** At the start of each step, the intra-step memory is cleared to prevent contamination from the previous state (Line 14). The `ExtractKV` function (Line 5, corresponding to Eq. 7) is invoked to extract KV pairs from the current observation $\hat{o}_{t-1}$, which are then stored in $\mathcal{M}_t^{\text{intra}}$ (Lines 15-16). This new intra-step memory is then fused with the historical cross-step memory $\mathcal{M}_t^{\text{cross}}$ using the `Merge` function (Line 19), which encapsulates the spatio-temporal fusion logic from Eqs. 8, 9, and 10. At the first step ($t = 1$), when $\mathcal{M}_t^{\text{cross}}$ is empty, the fused memory $\tilde{\mathcal{M}}_t$ is simply the intra-step memory (Line 18). Finally, the model predicts the action $\hat{a}_t$ using an enhanced attention mechanism conditioned on the fused memory $\tilde{\mathcal{M}}_t$, as described in Eq. 11 (Line 21).

**Navigation Visualization.** Following action prediction, the model generates the next observation $\hat{o}_t$. This process reuses the same fused memory $\tilde{\mathcal{M}}_t$ from the action prediction substep, ensuring contextual consistency. The generation is conditioned on the prior state and the newly predicted action $\hat{a}_t$ (Line 23). After both substeps, the intra-step memory $\mathcal{M}_t^{\text{intra}}$ is appended to the cross-step bank $\mathcal{M}_t^{\text{cross}}$, preserving the context of the current step for future predictions (Line 24).

This iterative process continues until the trajectory concludes, at which point the algorithm returns the complete sequences of predicted actions and observations (Line 27).

**Action Prediction**

**Input**

**Task:** Action Prediction
**Description:** Based on the current first-person observation, starting point observation and coordinate, goal point observation, predict the next action to take.
**Inputs:**
Starting Pose: (-90.16528149, -187.79242581, 0.15229973)

Start observation: 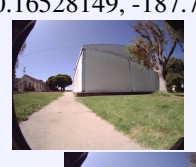    Goal observation: 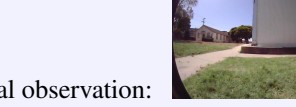

Current observation:
**Action Format:** The action can be the language command 'Stop', indicating the end of the trajectory. Alternatively, the action can be shifts composed of three components: - dx: displacement along the agent's facing direction), - dy: displacement perpendicular to the facing direction), - dyaw: change in heading angle (i.e., how much the agent rotates). All components are discretized into bin tokens: for example, - 'dx pos bin 02': dx = +0.02 meters, - 'dy neg bin 23': dy = -0.23 meters, - 'dyaw pos bin 26': counterclockwise rotation of +0.26 radians. If the agent reaches the goal or believes it has reached, it should predict 'Stop'. -Output format: Move by dx: <dx>, dy: <dy>, dyaw: <dyaw>
**Goal:** Predict the next action to approach the goal observation.

**Response**

**Predicted Action:**
Move by dx: <dx_pos_bin_18>, dy: <dy_pos_bin_05>, dyaw: <dyaw_pos_bin_07>

Figure 7: Prompt design details and examples on action prediction (context size = 1)

## C EXPERIMENTS AND RESULTS

### C.1 EVALUATION METRIC DETAILS

We evaluate overall system performance using two complementary categories of metrics:

**Navigation Quality:** For goal-conditioned visual navigation performance, the **Success Rate (SR)** defines a trajectory as successful if its final distance $d$ to the goal is smaller than the agent's average step size $\bar{s}$ (in meters). Formally, for trajectory $i$ among $N$ trajectories, with terminal estimate $\hat{p}_T^{(i)}$ and goal position $p_g^{(i)}$, SR is computed as

$$\text{SR} = \frac{1}{N} \sum_{i=1}^{N} \mathbf{1}\left[ d\left( \hat{p}_T^{(i)}, p_g^{(i)} \right) < \bar{s} \right],$$

**Absolute Trajectory Error (ATE)** quantifies global trajectory accuracy by measuring the Euclidean distance between aligned points of the predicted and reference trajectories. **Relative Pose Error (RPE)** instead captures local consistency, computed as the deviation in relative motion between successive estimated and ground-truth poses Sturm et al. (2012).

**(2) Visualization Quality:** For navigation visualization, visual predictions are evaluated with a combination of standard structural and perceptual measures, namely **SSIM** Wang et al. (2004), **PSNR** Hore & Ziou (2010), **LPIPS** Zhang et al. (2018), and **DreamSim** Fu et al. (2023). The

---

**Navigation Visualization**

Input

**Task:** Navigation Single Step Visualization
**Description:** Given the current first-person observation, predict the next first-person view observation after the agent executes a specified navigation action. To assist your prediction, you may refer to the start observation and pose (position: x, y and heading: yaw), as well as the goal and current observation.
**Inputs:**
Next Action: Move by dx: 0.18, dy: 0.05, dyaw: 0.07
Starting Pose: (-90.16528149, -187.79242581, 0.15229973)

Start Observation:     Goal Observation: 

Current Observation: 

**Action Format:** The action can be the language command 'Stop', indicating the end of the trajectory. Alternatively, the action can be shifts composed of three components: - dx: displacement along the agent's facing direction), - dy: displacement perpendicular to the facing direction), - dyaw: change in heading angle (i.e., how much the agent rotates). All components are discretized into bin tokens: for example, - 'dx pos bin 02': dx = +0.02 meters, - 'dy neg bin 23': dy = -0.23 meters, - 'dyaw pos bin 26': counterclockwise rotation of +0.26 radians.
**Spatial Interpretation:** - The magnitude of [dx, dy] reflects how far the agent moves in this step — larger values indicate greater positional shift, leading to larger visual changes. - dyaw controls the agent's rotation (change in heading). A positive dyaw indicates a left turn (counter-clockwise), while a negative dyaw indicates a right turn (clockwise).
**Goal:** Predict the most likely next first-person observation, considering how the movement and rotation implied by 'dx', 'dy', and 'dyaw' would affect what the agent sees next.

**Response**

**Predicted observation:** 

Figure 8: Prompt design details and examples on navigation visualization (context size = 1)

latter two are deep perceptual metrics specifically designed to more closely approximate human judgments. To assess longer-horizon stability under rollout, we introduce four metrics: **SSIM**@$n$, **PSNR**@$n$, **LPIPS**@$n$ and **DreamSim**@$n$. Standard one-step metrics compare ground-truth next frame $o_{t+1}$ with one-step prediction $\hat{o}_{t+1}^{(1)}$ obtained from ground truth current observation and action $(o_t, a_{t+1})$. For horizon $n$, we perform open-loop rollout that recursively feeds the model's predicted observations back as inputs while conditioning on ground-truth action sequence $a_{t+1:t+n+1}$: SSIM@$n$ = SSIM$\big(o_{t+n}, \hat{o}_{t+n}^{(n)}\big)$, where $\hat{o}_{t+n}^{(n)}$ is the observation prediction after $n$ rollouts, with PSNR@$n$, LPIPS@$n$ and DreamSim@$n$ defined analogously by replacing SSIM with the corresponding measure. We also provide detailed calculations for LPIPS and DreamSim here.

**LPIPS:** The Learned Perceptual Image Patch Similarity quantifies perceptual resemblance by computing weighted distances between deep feature activations extracted from pretrained vision back-

---

**Algorithm 1** Inference with Intra-step and Cross-step Memory Banks in UniWM

---

**Input:** Start position $p_0$, start observation $o_s$, goal observation $o_g$; Decoder layers $L = \{l_0, \ldots, l_{31}\}$
**Output:** Action sequence $A_T = \{\hat{a}_1, \ldots, \hat{a}_T\}$, observation sequence $\mathcal{O}_T = \{\hat{o}_1, \ldots, \hat{o}_T\}$
  1: **Definitions (helpers)**
  2:   $\texttt{ResetIntra}()$: clear intra-step memory bank $\mathcal{M}_t^{\text{intra}}$
  3:   $\texttt{AppendIntra}(\{K_t^{(l)}, V_t^{(l)}\}_{l \in L_{\text{save}}})$: push layer-wise KV to $\mathcal{M}_t^{\text{intra}}$
  4:   $\texttt{AppendCross}(\mathcal{M}_t^{\text{intra}})$: push intra-step bank $\mathcal{M}_t^{\text{intra}}$ to cross-step bank $\mathcal{M}^{\text{cross}}$
  5:   $\texttt{ExtractKV}(\text{token seq.}) \mapsto \{K_t^{(l)}, V_t^{(l)}\}_{l \in L_{\text{save}}}$: extract KV at selected layers (Eq. 7)
  6:   $\texttt{Merge}(\mathcal{M}_t^{\text{cross}}, \mathcal{M}_t^{\text{intra}}) \mapsto \tilde{\mathcal{M}}_t$: memory fusion (Eqs. 8, 9, and 10)
  7:   $\texttt{EnhanceAndDecode}(\text{cond}, \mathcal{M}_t^{\text{intra}}, \tilde{\mathcal{M}}_t) \mapsto$ predict with enhanced attention (Eq. 11)
  8: **Initialization**
  9: $\mathcal{M}_t^{\text{intra}} \leftarrow \varnothing, \quad \mathcal{M}_t^{\text{cross}} \leftarrow \varnothing$  $\triangleright$ cross-step memory is persistent across steps
 10: $\hat{o}_0 \leftarrow o_s, \quad L_{\text{save}} \leftarrow \{l_0, l_7, l_{15}, l_{23}, l_{31}\}$
 11: **for** $t = 1$ **to** $T$ **do**
 12:   $\texttt{ResetIntra}()$  $\triangleright$ always reset intra-step memory at a new step
 13:   $\{K_t^{(l)}, V_t^{(l)}\} \leftarrow \texttt{ExtractKV}(p_0, o_s, o_g, \hat{o}_{t-1})$
 14:   $\texttt{AppendIntra}(\{K_t^{(l)}, V_t^{(l)}\}_{l \in L_{\text{save}}})$
 15:   **Substep A: Action prediction at step** $t$
 16:   **if** $\mathcal{M}_t^{\text{cross}} = \varnothing$ **then**
 17:     $\tilde{\mathcal{M}}_t \leftarrow \mathcal{M}_t^{\text{intra}}$  $\triangleright$ no cross memory at $t=1$
 18:   **else**
 19:     $\tilde{\mathcal{M}}_t \leftarrow \texttt{Merge}(\mathcal{M}_t^{\text{cross}}, \mathcal{M}_t^{\text{intra}})$
 20:   **end if**
 21:   $\hat{a}_t \leftarrow \texttt{EnhanceAndDecode}((p_0, o_s, o_g, \hat{o}_{t-1}), \tilde{\mathcal{M}}_t)$
 22:   **Substep B: Navigation Visualization at step** $t$
 23:   $\hat{o}_t \leftarrow \texttt{EnhanceAndDecode}((p_0, o_s, o_g, \hat{o}_{t-1}, \hat{a}_t), \tilde{\mathcal{M}}_t)$
 24:   $\texttt{AppendCross}(\mathcal{M}_t^{\text{intra}})$  $\triangleright$ Deposit intra-step memory to cross-step memory
 25: **end for**
 26: **return** $A_T = \{\hat{a}_1, \ldots, \hat{a}_T\}, \mathcal{O}_T = \{\hat{o}_1, \ldots, \hat{o}_T\}$

---

bones (e.g., AlexNet, VGG). By operating in a learned feature space, LPIPS better captures perceptually relevant differences than conventional low-level pixel-level measures.

**DreamSim:** DreamSim extends perceptual evaluation to the multimodal domain by measuring semantic alignment between generated images and a target text description. Given images $\{I_i\}_{i=1}^N$ and a prompt $T$, it is defined as:

$$\text{DreamSim}(I_{1:N}, T) = \frac{1}{N} \sum_{i=1}^N \frac{\langle f_{\text{img}}(I_i), f_{\text{text}}(T) \rangle}{\|f_{\text{img}}(I_i)\| \cdot \|f_{\text{text}}(T)\|}. \tag{12}$$

DreamSim leverages fused or fine-tuned visual–textual features (e.g., CLIP, OpenCLIP, DINO) trained on synthetic human similarity judgments, thereby further enhancing sensitivity to nuanced perceptual and semantic correspondences. By combining LPIPS and DreamSim, our evaluation jointly accounts for low-level visual fidelity and high-level semantic coherence, offering a balanced and human-aligned assessment across both structural and semantic dimensions.

## C.2 Inference Time

We provide a comparison of average inference time per trajectory together with navigation metrics across the four datasets (Go Stanford, ReCon, SCAND, HuRoN).

As shown in Table 9, NoMaD achieves fast inference but lacks imagination capability, which limits success rates in challenging cases. World-model-based approaches (NWM, Anole-7B, UniWM) incur higher inference costs due to visual imagination. Importantly, UniWM runs substantially faster than its backbone Anole-7B and NWM, while delivering markedly better navigation performance, demonstrating a favorable balance between efficiency and accuracy. Quantization to 4-bit Frantar et al. (2022) can potentially power UniWM up to an average of 16s per trajectory.

Table 9: **Comparison of average inference time per trajectory** together with navigation metrics (SR, ATE, RPE) across the four datasets (Go Stanford, ReCon, SCAND, HuRoN)

| Method | Avg SR | Avg ATE | Avg RPE | Avg Inference Time per Trajectory (second) |
|---|---|---|---|---|
| NoMaD Sridhar et al. (2024) | 0.48 | 0.79 | 0.26 | 0.6 |
| NWM Bar et al. (2025) | 0.55 | 0.63 | 0.23 | 114×32 |
| Anole-7B Chern et al. (2024) | 0.27 | 1.80 | 0.73 | 654 |
| UniWM (with Memory) | 0.78 | 0.32 | 0.12 | 82 |
| UniWM + Quant. 4-bit (with Memory) | - | - | - | 16 (est. Frantar et al. (2022)) |

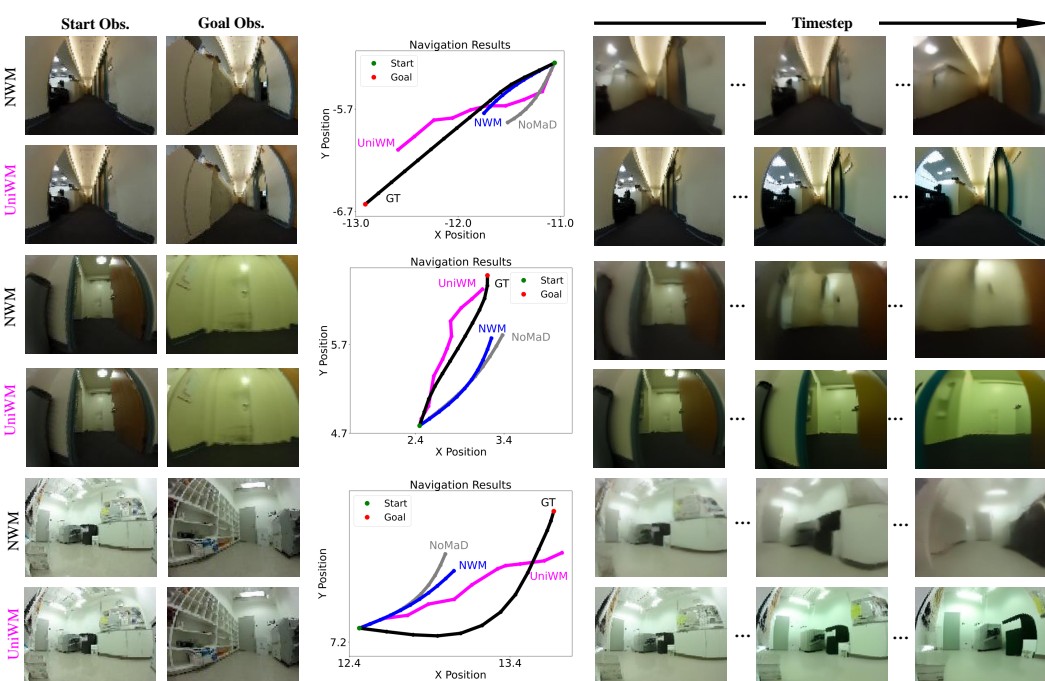

Figure 9: **Qualitative Comparisons** on Go Stanford across UniWM, NWM, and NoMaD. The central trajectory plots highlight difference between predicted $A_T$ and the ground-truth.

## C.3 MORE QUALITATIVE RESULTS

We provide more qualitative results in Figs. 9, 10 and 11.

## D USE OF LLMS

Large Language Models (LLMs) were used to aid in the writing and polishing of the manuscript. Specifically, we used an LLM to assist in refining the language, improving readability, and ensuring clarity in various sections of the paper. The model helped with tasks such as sentence rephrasing, grammar checking, and enhancing the overall flow of the text.

It is important to note that the LLM was not involved in the ideation, research methodology, or experimental design. All research concepts, ideas, and analyses were developed and conducted by the authors. The contributions of the LLM were solely focused on improving the linguistic quality of the paper, with no involvement in the scientific content or data analysis.

The authors take full responsibility for the content of the manuscript, including any text generated or polished by the LLM. We have ensured that the LLM-generated text adheres to ethical guidelines and does not contribute to plagiarism or scientific misconduct.

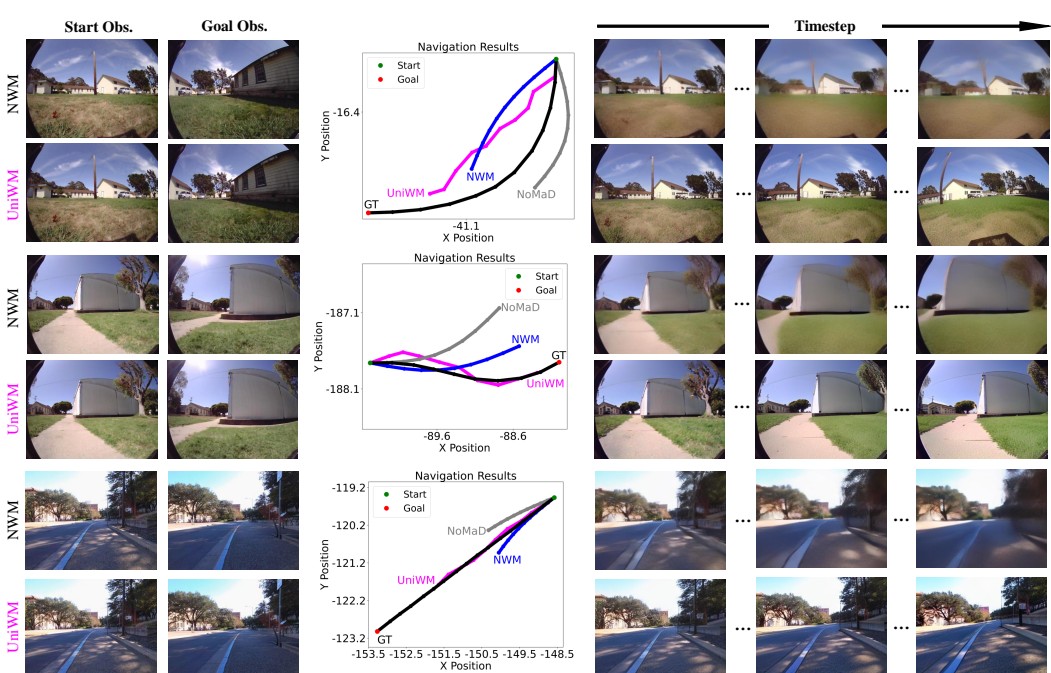

Figure 10: **Qualitative Comparisons** on ReCon and Scand across UniWM, NWM, and NoMaD. The central trajectory plots highlight difference between predicted $A_T$ and the ground-truth.

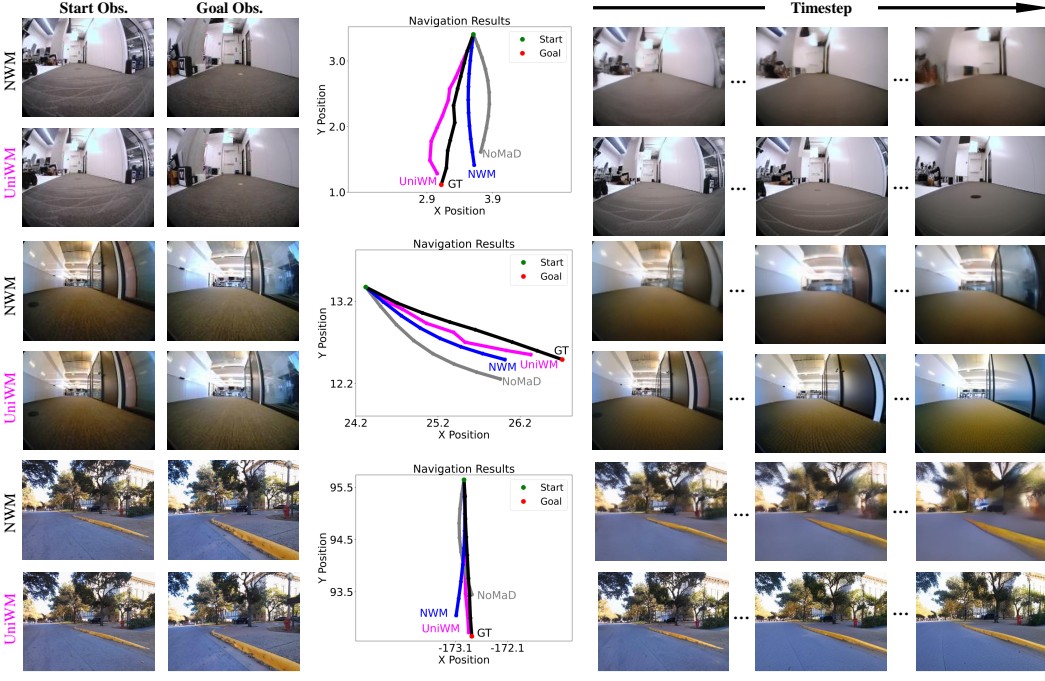

Figure 11: **Qualitative Comparisons** on HuRoN across UniWM, NWM, and NoMaD. The central trajectory plots highlight difference between predicted $A_T$ and the ground-truth.

