# OpenReview forum: "Unified World Models: Memory-Augmented Planning and Foresight for Visual Navigation"
_ICLR.cc/2026/Conference — Submitted to ICLR 2026_

### Official Review · Reviewer_LfkC · 2025-10-21

**Soundness:** 3
**Presentation:** 3
**Contribution:** 2
**Rating:** 4
**Confidence:** 4

**Summary:**

This paper proposes UniWM, a unified multimodal autoregressive world model for goal-conditioned visual navigation. Unlike prior modular approaches that decouple planning and imagination, UniWM integrates both within a single backbone and augments inference with a hierarchical memory mechanism. The model alternates between predicting actions and visualizing future observations, leveraging intra-step and cross-step memory banks to stabilize long-horizon rollouts. Extensive experiments across four benchmarks and zero-shot generalization to TartanDrive show strong performance gains over existing baselines.

**Strengths:**

- **Clarity and Motivation**: The paper is well-written, with a clear problem framing and motivation. It effectively critiques limitations of prior modular designs and positions UniWM as a principled alternative.

- **Unified Architecture**: The integration of planning and imagination into a single autoregressive backbone is novel and addresses state–action misalignment, which is also validated in the ablation study.

- **Memory-Augmented Inference**: The hierarchical memory mechanism (KV-cache based) is well-motivated and empirically shown to improve long-horizon stability.

- **Comprehensive Experiments**: UniWM outperforms baselines across multiple datasets in both navigation and visualization metrics. Ablation studies are thorough and isolate contributions of memory, tokenization, and training strategies.

**Weaknesses:**

1.  **Problem Formulation Violates MDP Principles**:

    The world model is defined as $o_{t+1} = W(o_t, a_{t+1}, o_s, o_g)$, which violates the standard MDP formulation. In MDPs, the transition model should **only reflect environment dynamics** and **be independent of the task goal $o_g$**. Conditioning on $o_g$ risks conflating planning with dynamics modeling, and this theoretical inconsistency undermines the model’s interpretability and generalizability.

2. **Information Leakage via Goal Conditioning**:

    Conditioning the **transition model** on the final **goal observation $o_g$** introduces potential leakage, especially in **visualization metrics** like SSIM and LPIPS. This is particularly problematic for **state-action pairs near the goal**, where the model may trivially reconstruct the goal view. **Comparisons with NWM (which does not use $o_g$ during prediction) are  thus not entirely fair**. Additional experiments isolating the impact of $o_g$ conditioning —e.g., removing it or masking it—would strengthen the claims.

3. **Unfair Dataset Usage in Benchmark Comparisons**:

    The paper reports large gains on **Go Stanford**, but unlike NWM (Go stanford is only used for evaluation), UniWM uses this dataset during training. This undermines the fairness of the comparison and inflates the perceived contribution.

4. **Training Details and Architectural Transparency**:
    Several aspects of the training pipeline are under-specified:
    - The label smoothing loss $L_{LS}$ is mentioned but not clearly differentiated from the main objectives.
    - The implementation of cross-attention between the decoder and memory modules is not described. Did the authors modify standard Transformer blocks or insert new cross-attention layers between the decoder and memory module?

**Questions:**

1. Discretized Bin Token Loss: Why was this chosen over naive cross-entropy?

---

> ### Author Response · Authors · 2025-11-20
>
> **W1:** We appreciate the reviewer’s observation that, under the strict MDP definition, transition dynamics should be independent of the task goal. Our formulation is not intended to redefine environment dynamics, but rather to **instantiate a goal‑conditioned generative world model tailored for visual navigation**. In partially observable settings, conditioning on the goal observation is essential to reduce multimodal uncertainty and to align imagined trajectories with task requirements. Thus, our contribution should be interpreted not as a violation of MDP principles, but as **a practical extension of the MDP framework to goal‑conditioned navigation**, where the agent must imagine task‑relevant futures rather than average environment dynamics. We will clarify this distinction in the revised manuscript.
>
> To further demonstrate that conditioning on the goal image does not compromise generalization, we conducted an additional experiment. We retrain UniWM from scratch on the same four datasets, but **remove goal image $o_{g}$ from the navigation visualization substep during both training and inference**, keeping all other settings unchanged. As shown in Table below (**we evaluate on the unseen TartanDrive split**), the original UniWM still **outperforms** the retrained variant in unseen environments, indicating that conditioning on the **goal image does not harm generalization** in our task.
>
>  | Method                        | SR ↑ | ATE ↓ | RPE ↓ |
> |-------------------------------|------|-------|-------|
> | UniWM (w/o $o_g$ in navigation visualization substep (retrained))  | 0.33 | 1.37  | 0.51  |
> | UniWM (origin)           | **0.35** | **1.20**  | **0.46**  |
>
> **W2:** We thank the reviewer for raising the concern about potential information leakage through goal conditioning. To directly address this, we conduct additional experiments that isolate the impact of conditioning on the goal observation. Specifically, we **remove goal image $o_g$ from the navigation visualization substep during inference**, while keeping all other settings unchanged.
>
> As shown in the table below, even when we remove goal image $o_g$ at inference, **UniWM still achieves strong performance and remains better than NWM**. This shows that our conclusions do not rely on a special advantage from goal conditioning, and the comparison remains fair. And the original UniWM consistently outperforms the variant without goal conditioning across both navigation and imagination metrics. This demonstrates that conditioning on the goal image does not simply inflate visualization scores, but instead provides meaningful task context that helps align imagined rollouts with navigation objectives.
>
>  | Method                        | SR ↑ | ATE ↓ | RPE ↓ | SSIM ↑ | LPIPS ↓ | DreamSim ↓ | SSIM@5 ↑ | LPIPS@5 ↓ | DreamSim@5 ↓ |
> |-------------------------------|------|-------|-------|--------|---------|------------|----------|-----------|--------------|
> | NWM  |  0.55   | 0.63    | 0.23 | 0.389  | 0.318   | 0.089      | 0.256    | 0.494    | 0.174       |
> | UniWM (remove $o_g$ in navigation visualization substep during inference)  | 0.68 | 0.39  | 0.15  | 0.427  | 0.293   | 0.058      | 0.332    | 0.476     | 0.160        |
> | UniWM (origin)           | **0.71** | **0.36**  | **0.13**  | **0.457**  | **0.254**   | **0.041**      | **0.350**    | **0.435**     | **0.126**        |
>
> **W3:** For fairness and we did not get the access to the official NWM checkpoints, we **retrained all diffusion-based baselines, including NWM, from scratch using the same four training datasets as UniWM (Go Stanford, ReCon, SCAND, HuRoN), and evaluated zero-shot on TartanDrive**. Thus, all baselines were trained under **identical conditions**. On the other hand, we chose TartanDrive as the out-of-domain evaluation dataset because it uniquely contains robot body structures in observations that evolve during navigation, making it a more realistic OOD benchmark as noted in sec.3.3. We believe **this setting better reflects real-world distribution shift**.

---

> > ### Author Response · Authors · 2025-11-20
> >
> > **W4:** We thank the reviewer for pointing out the need for more clarity in the training pipeline and architectural details.
> >
> > **On label smoothing loss.**
> > Label smoothing is **used only in ablation experiments (Fig. 5) as a baseline, not in our final training recipe**. Specifically: For action tokens, our main loss is the discretized bin-token loss $L_{\text{plan}}$. In the ablation, we compare this against a label-smoothing cross-entropy baseline $L_{\text{LS}}$ applied to action tokens. For image tokens, our main loss is the reconstruction-aware loss $L_{\text{world}}$. In the ablation, we compare this against label smoothing $L_{\text{LS}}$ applied to predicted image tokens. Thus, in Fig. 5: The bar labeled $L_{\text{LS}} + L_{\text{LS}}$ uses label smoothing for both action and image tokens. The bar labeled $L_{\text{plan}} + L_{\text{LS}}$ uses our bin-token loss for actions and label smoothing for images, etc.
> >
> > We will make this explicit in the training section to clearly distinguish the main objectives from the ablation baselines.
> >
> > **On cross‑attention with memory.**
> > We did not conduct architectural modifications on the Transformer blocks or insert new cross-attention layers. Instead, we **augment the standard decoder cross‑attention by supplying fused memory key–value states**, as described in Eq.11. At each step, the intra‑step memory $M_{\text{intra}}$ and cross‑step memory $M_{\text{cross}}$ are fused via similarity gating and temporal decay (Eqs.8–10). The resulting memory states are concatenated with the current observation tokens and passed into the decoder’s existing cross‑attention. Algorithm 1 in the paper explicitly outlines this rollout procedure: at each step, the intra‑step memory is reset, fused with historical memory, and then injected into the decoder’s cross‑attention to guide both the planner and visualization substeps. We will expand Sec.2.3 and the description around Algorithm 1.
> >
> > **Q1:** We thank the reviewer for the question regarding our choice of discretized bin‑token loss. As shown in the table below, we conducted a direct comparison between using a naive cross‑entropy loss $L_{CE}$ to optimize all tokens in the action prediction substep versus our proposed bin‑token loss $L_\text{plan}$. In both cases, the navigation visualization substep was trained with the reconstruction loss $L_\text{world}$. From these results, it is clear that **the bin‑token formulation yields substantially higher success rate and lower trajectory errors compared to naive cross‑entropy**. This confirms that discretizing continuous actions into structured bin tokens provides a more effective training signal for navigation planning, and validates our choice of $L_\text{plan}$ as the primary objective for action prediction.
> >
> >
> > | Loss for Action Tokens        | Loss for Image Tokens        | SR   | ATE  | RPE  |
> > |-------------------------------|------------------------------|------|------|------|
> > | $L_{CE}$            | $L_\text{world}$           | 0.57 | 0.51 | 0.22 |
> > | $L_\text{plan}$   | $L_\text{world}$           | **0.71** | **0.36** | **0.13** |

---

> ### Comment · Reviewer_LfkC · 2025-11-28
>
> Thank the authors for the rebuttal, which addresses some of my concerns regarding training data usage and includes an additional experiment on the effect of goal conditioning.
>
> However, my primary concern regarding methodological rigor remains. While I do not oppose the motivation of unifying the transition model and planner within a shared backbone, **conditioning the transition model on the goal image seriously violates the principles of the MDP formulation**. This issue could lead to **significant prediction bias (misled by the goal) in cases where the goal image is misleading**.
>
> I believe this work has potential to be compelling. I would suggest a methodological restructuring (e.g., maintaining the shared architecture but using techniques like masking to isolate unnecessary information between the planner and transition model during inference).
>
> Given that the core conceptual issue persists, I maintain my original rating of 4 (Weak Reject).

---

> ### Author Response · Authors · 2025-12-01
>
> We are happy to have **already addressed your concerns regarding training data usage and the effect of goal conditioning**, and we sincerely **appreciate your recognition of our motivation to unify the transition model and planner within a shared backbone**.
>
> Regarding the reviewer’s concern that conditioning on the goal image may affect generalization or introduce potential leakage, it seems that this concern persists **despite the evidence provided**. We would like to **reiterate our clarifications and supporting evidence**.
>
> **Clarification on MDP principles**:
>
> Our formulation is not intended to redefine environment dynamics, but rather to instantiate a goal‑conditioned generative world model tailored for visual navigation. In partially observable settings, conditioning on the goal observation is essential to reduce multimodal uncertainty and align imagined trajectories with task requirements. Thus, our contribution should be interpreted not as a violation of MDP principles, but as **a practical extension of the MDP framework to goal‑conditioned navigation**, where the agent imagines task‑relevant futures rather than average dynamics.
>
> **Additional experiments**:
>
> To directly test the reviewer’s concern, we retrain and evaluate UniWM in unseen environments with the goal image removed during both training and inference in the first table below, and also evaluate origin UniWM in seen environments with the goal image removed only during inference, while keeping all other settings unchanged (the second table below). Across both settings, **the original UniWM consistently outperforms the variants without goal conditioning**, and remains stronger than NWM. These results demonstrate that conditioning on the goal image does not harm unseen environment generalization, nor does it simply inflate visualization scores. Instead, **it provides meaningful task context** that helps align imagined rollouts with navigation objectives.
>
>  | Method                        | SR ↑ | ATE ↓ | RPE ↓ |
> |-------------------------------|------|-------|-------|
> | UniWM (w/o $o_g$ in navigation visualization substep (retrained))  | 0.33 | 1.37  | 0.51  |
> | UniWM (origin)           | **0.35** | **1.20**  | **0.46**  |
>
>  | Method                        | SR ↑ | ATE ↓ | RPE ↓ | SSIM ↑ | LPIPS ↓ | DreamSim ↓ | SSIM@5 ↑ | LPIPS@5 ↓ | DreamSim@5 ↓ |
> |-------------------------------|------|-------|-------|--------|---------|------------|----------|-----------|--------------|
> | NWM  |  0.55   | 0.63    | 0.23 | 0.389  | 0.318   | 0.089      | 0.256    | 0.494    | 0.174       |
> | UniWM (remove $o_g$ in navigation visualization substep during inference)  | 0.68 | 0.39  | 0.15  | 0.427  | 0.293   | 0.058      | 0.332    | 0.476     | 0.160        |
> | UniWM (origin)           | **0.71** | **0.36**  | **0.13**  | **0.457**  | **0.254**   | **0.041**      | **0.350**    | **0.435**     | **0.126**        |
>
> We emphasize that **these clarifications and additional experiments were conducted precisely in response to the reviewer’s feedback, and the results consistently support our interpretation**. While we respect the reviewer’s perspective, it seems that the concern about prediction bias persists **despite the evidence provided**. We will further highlight these clarifications and results in the revised manuscript.

---

### Official Review · Reviewer_fGwF · 2025-10-30

**Soundness:** 2
**Presentation:** 3
**Contribution:** 2
**Rating:** 2
**Confidence:** 4

**Summary:**

This paper proposes a unified architecture for policy and world model, and a plug-in hierarchical memory mechanism using KV caches. The proposed method is evaluated on goal -conditioned navigation tasks and shows competitive performance compared to NWM and other baselines. However, the reported results are questionable given the discrepancies in training settings with the critical baseline. I will be willing to increase my score if my concerns regarding the experiment setting are addressed.

**Strengths:**

1. The paper is well written and easy to follow.
2. The proposed hierarchical memory mechanism is intriguing and shows some good potential for navigation tasks.

**Weaknesses:**

1. The unified architecture of predicting both action and future images has been used in multiple other works [1][2] so novelty in this aspect is limited. As the authors claim this unified architecture and joint training will help align imagination with control, it will be helpful to have some experiments to support this claim. For example, separate planner and world models can be trained with same training data and compared to this proposed unified model.

2. My primary concern is about the experiment setup. Clearly NWM is the strongest baseline that this work tries to beat, but there are some significant discrepancies in the experiment setup. NWM uses ReCon, SCAND, HuRoN and TartanDrive for training and in-domain evaluation, while using GO Stanford for OOD evaluation due to the low resolution images. However, this paper uses TartanDrive for OOD evaluation and the other four datasets for training. Is there any specific reason that you have to make this change? Did you train NWM yourself on the same training datasets as yours? Otherwise all the conclusions drawn from comparison to NWM's results are significantly compromised.

3. I notice that NWM reported ATE 5.63 and RPE 1.18 in their paper as in-domain evaluation, but the out-of-domain results in Table 6 of the paper (which should be harder) shows that NWM achieves an ATE of 1.61 and a RPE of 0.62, both of which are much better. How is NWM evaluated in your paper and is there any explanation for this discrepancy?

[1] Imagine while Reasoning in Space: Multimodal Visualization-of-Thought https://arxiv.org/abs/2501.07542
[2] WorldVLA: Towards Autoregressive Action World Model https://arxiv.org/abs/2506.21539

**Questions:**

1. Experiment result of separate planner and world models trained with same data.
2. It is quite surprising for me that the proposed hierarchical memory using KV caches works this well without any finetuning, considering they are from early layers (layer 5-7). Based on the description, this memory is used for both action and image generation. Do you know in which step (action or image generation) it has more significant impact? Do you think if this memory mechanism can be directly applied to other policy or world model networks?
3. Other questions are listed in the weakness session.

---

> ### Author Response · Authors · 2025-11-20
>
> **W1 & Q1:** We thank the reviewer for the detailed comments. We respond to each point below and clarify why we believe our experimental setup and claims are well-founded. We consider goal‑conditioned visual navigation to be an important and highly challenging task for embodied agents. Our work differs from [1] in several key aspects: while [1] introduces a multimodal reasoning mechanism, it restricts reasoning to **relatively narrow scenarios** such as 2D maze games, without addressing real‑world applications. Moreover, it overlooks the **critical role of memory in unifying planning and imagination**, an ability that is essential in realistic environments.
>
> Our work also differs from [2], which predicts long action sequence in a single step and ignores the benefits of intermediate reasoning and memory for planning. In contrast, our setting requires agents not only to **imagine while acting but also to remember over time**.
>
> UniWM is the **first framework that is both unified and memory‑augmented in goal‑conditioned visual navigation**, enabling stable long‑horizon rollouts. We agree that jointly predicting actions and future images is indeed a valuable research direction, and we appreciate the reviewer’s point. We will continue to follow developments in this area closely.
>
> To directly address the reviewer’s concern, we conduct an additional experiment where we **trained separate planner and world models using the same training data and schedule**, and compare them with our unified model in the table below. The unified UniWM clearly **outperforms** the separate planner + world-model setup on both navigation (SR, ATE, RPE) and visualization metrics (SSIM, LPIPS, DreamSim), which provides direct empirical evidence that our joint architecture and training indeed better align imagination with control.
>
> |Method|SR ↑|ATE ↓|RPE ↓|SSIM ↑|LPIPS ↓|DreamSim ↓|SSIM@5 ↑|LPIPS@5 ↓|DreamSim@5 ↓|
> |-|-|-|-|-|-|-|-|-|-|
> |UniWM (separate planner + world model) |0.65|0.41|0.16|0.443|0.280|0.055| 0.329    | 0.470| 0.154|
> | UniWM  (origin) | **0.71** | **0.36**  | **0.13**  | **0.457**| **0.254**| **0.041**| **0.350**| **0.435**| **0.126**|
>
>
> **W2:** We chose TartanDrive as the out-of-domain evaluation dataset because it uniquely contains **visible robot body structures and embodiment-related artifacts** (e.g., bumper/hood regions) that evolve over time. This makes TartanDrive a more demanding OOD benchmark for goal-conditioned visual navigation as noted in sec.3.3, where the goal is to generalize across embodiments rather than only across camera resolution. We believe this setting **better reflects real-world distribution shift**.
>
> For fairness, and because we did not have the access to the official NWM checkpoints, we **retrain all diffusion-based baselines, including NWM, from scratch using the same four training datasets as UniWM (Go Stanford, ReCon, SCAND, HuRoN), and evaluated zero-shot on TartanDrive**. Thus, UniWM and NWM (as well as other baselines) are trained under **identical conditions**, and any performance difference is not due to training data mismatch. We will explicitly state this training–evaluation configuration in the revised manuscript.
>
> **W3:** These results also surprise us. In our evaluation pipeline, NWM’s performance improved across datasets compared to its original report (see Table 1). We **strictly follow NWM’s best reported configuration**: context length = 4, goal state = 4, CDiT-XL backbone (1B parameters) with both action and time conditioning. Within our pipeline, NWM’s performance was consistent and served as a fair baseline for comparison.
>
> **Q2:** We appreciate the reviewer’s observation. Based on our analysis, the hierarchical memory mechanism has a **more pronounced impact on image generation** than on action prediction. This is because visual imagination requires maintaining spatial–temporal coherence across multiple steps, and the hierarchical memory provides rich perceptual embeddings that stabilize long‑horizon rollouts. By fusing intra‑step and cross‑step caches, the hierarchical memory ensures that both immediate perceptual cues and longer‑range trajectory context are preserved, yielding consistent improvements in navigation and visualization.
>
> To illustrate, we provide an extension of Table 4 averaged across four datasets, showing the impact of memory layer selection. As shown below, Layer 5 achieves the best balance across navigation and visualization metrics, confirming **our choice of early‑layer memory as both effective and principled**.
>
>
> | Layer Num | SR ↑ | ATE ↓ | RPE ↓ | SSIM ↑ | LPIPS ↓ | DreamSim ↓ |
> |-|-|-|-|-|-|-|
> |1| 0.72 | 0.36  | 0.13  | 0.462  | 0.250| 0.041|
> |3| 0.76 | 0.34  | **0.12**|0.468|0.247|0.040|
> |5| **0.78** | **0.32**  | **0.12**  | **0.473** | **0.239** | 0.040 |
> |7| 0.77 | **0.32**  | **0.12**  | 0.471  | 0.243   | **0.038** |
> |16|0.60|0.52|0.24|0.365|0.342|0.089|
> |32|0.57|0.58|0.27|0.341|0.368|0.095|

---

> > ### Author Response · Authors · 2025-11-20
> >
> > **Reference**
> >
> > [1] Imagine while Reasoning in Space: Multimodal Visualization-of-Thought https://arxiv.org/abs/2501.07542
> >
> > [2] WorldVLA: Towards Autoregressive Action World Model https://arxiv.org/abs/2506.21539

---

> > ### Comment · Reviewer_fGwF · 2025-11-24
> > **Clarification of training details for NWM baseline**
> >
> > A follow-up question regarding your responses to W2 and W3. You said you re-trained NWM by "strictly follow NWM’s best reported configuration" using the same dataset as UniWM. Did you use the official repo of NWM or the training script you developed? Are the data processing script and evaluation pipeline aligned with NWM official repo?

---

> > > ### Author Response · Authors · 2025-11-28
> > >
> > > We thank the reviewer for raising this important clarification. For training and evaluation, we used the **official NWM repository and strictly followed their best reported configuration** (context length = 4, goal state = 4, CDiT‑XL backbone with ~1B parameters, including both action and time conditioning) to train NWM **from scratch on the same four datasets** (Go Stanford, ReCon, SCAND, and HuRoN) as UniWM. We also used **the same calculation** for all common evaluation metrics between our work and NWM (e.g., ATE, RPE, LPIPS, and DreamSim).
> > >
> > > Regarding datasets, as noted in the NWM repository, **no dataset is directly provided**; instead, they instruct users to “To download and preprocess data, please follow the steps from NoMaD.” Accordingly, we adopted the NoMaD preprocessing scripts to prepare all five datasets used in our paper (Go Stanford, ReCon, SCAND, HuRoN, and TartanDrive). After completing all preprocessing steps exactly as specified, we observed that approximately 23.4% of trajectories contained fewer than three egocentric observations, and about 14.9% contained only a single egocentric observation. This implies that **for ~14.9% of trajectories, no valid goal‑conditioned navigation path exists**, and **for roughly one quarter of trajectories, the start observation alone suffices to reach the goal in a single step**, leaving no meaningful intermediate navigation process. To ensure the validity of the goal‑conditioned navigation task, we therefore applied an additional filtering step beyond the official NoMaD/NWM preprocessing, **filtering out all trajectories shorter than three steps**, as described in **Section 3 (Datasets)**.
> > >
> > > To further study the impact of these extremely short trajectories on navigation performance, we conduct an additional experiment where we retain all trajectories including those shorter than three steps, and train both NWM and UniWM from scratch on the same four datasets (Go Stanford, ReCon, SCAND, and HuRoN) without filtering. All other settings are kept identical, and we report the averaged performance across these four datasets as in table below. Even under this setting, UniWM consistently **outperforms** NWM in goal‑conditioned visual navigation. **We hope that all the clarifications, additional experiments, and the efforts presented here sufficiently address the reviewer’s concerns**.
> > >
> > > | Method | SR ↑  | ATE ↓ | RPE ↓ |
> > > |--------|-------|-------|-------|
> > > | NWM    | 0.31  | 1.05  | 0.47  |
> > > | UniWM  | **0.58**  | **0.49**  | **0.24**  |

---

### Official Review · Reviewer_zu2f · 2025-10-31

**Soundness:** 3
**Presentation:** 3
**Contribution:** 3
**Rating:** 6
**Confidence:** 4

**Summary:**

This paper proposes UniWM, a unified, memory-augmented world model trying to address critical limitations in existing visual navigation.
UniWM integrates egocentric visual foresight and navigation planning into a single multimodal autoregressive backbone, explicitly grounding action decisions in visually imagined outcomes to ensure prediction-control alignment. This paper introduces a hierarchical memory mechanism, where intra-step memory caches immediate perceptual cues, while cross-step memory accumulates long-term trajectory context. Experiments on four benchmarks (Go Stanford, ReCon, SCAND, HuRoN) show UniWM boosts navigation success rates (SR) by up to 30% and reduces trajectory errors (ATE/RPE) versus baselines (e.g., GNM, VINT, NoMaD, NWM).

**Strengths:**

(1) By bridging visual imagination and navigation planning into a single multimodal autoregressive backbone, this paper demonstrates that both tasks can be mutually beneficial to tighten alignment between prediction and control, which sets a promising scheme for future visual navigation approaches.

(2) The proposed hierarchical memory bank mechanism is an interesting and efficient idea to incorporate historical inputs into the visual planning and imagination process. By reusing the key-value (KV)-cache stored in selected decoder layers (intra-step memory) and across temporal steps (cross-step memory), the scheme avoids redundant computation of past perceptual cues and trajectory context.

(3) All the proposed components are fairly evaluated and analyzed with clear ablation studies, making it easy to quantify and understand the technical contribution of each part of the paper.

(4) I appreciate the authors’ efforts to support reproducibility, including the code in the supplementary materials

**Weaknesses:**

(1) The planning results are evaluated within an open-loop evaluation protocol, which limits the ability to fully assess UniWM’s long-horizon performance and generalization.

(2) The demonstrated results do not include scenarios with dynamic obstacles, a critical gap given that dynamic elements (e.g., moving pedestrians, vehicles) are common and challenging in real-world navigation tasks.

(3) The paper lacks efficiency analysis compared to state-of-the-art navigation methods like NoMaD—an oversight given that real-time planning frequency is a critical requirement for visual navigation tasks.

**Questions:**

(1) The paper states that UniWM is fine-tuned on the GAIR Anole-7B backbone, but the rationale for selecting Anole-7B specifically remains unclear. Could the authors elaborate on the key factors that led to the choice of Anole-7B over other existing multimodal large language models (MLLMs) (e.g., Qwen-VL-2.5) for visual navigation tasks?

(2) There are other world models also support imagining the planning process given start and target images—aligning with UniWM’s goal of integrating visual imagination and navigation planning. How does the UniWM performance compares with the Aether in the navigation benchmarks?

(3) The paper emphasizes UniWM’s practical value for real-world navigation where inference speed directly impacts deployment viability. However, it provides no details on the time cost of decoding entire navigation trajectories.  Could the authors specify the average inference time per trajectory on the evaluated datasets?

[1] Team, Aether, et al. "Aether: Geometric-aware unified world modeling." arXiv preprint arXiv:2503.18945 (2025).

---

> ### Author Response · Authors · 2025-11-20
>
> **W1:** We appreciate the reviewer’s concern regarding the use of an open‑loop evaluation protocol. We adopted this protocol because it is the **established standard** for goal‑conditioned visual navigation, consistently used by prior baselines such as GNM, VINT, NoMaD, and NWM. This choice ensures **fair comparability** across methods. At the same time, we agree that extending UniWM to closed‑loop evaluation is an important next step.
>
> It is also important to emphasize that open‑loop evaluation is not unique to navigation. **Similar protocols** are widely applied in autonomous driving [1][2], robotic manipulation [3][4], and gaming [5], where long‑horizon reasoning must be assessed without the confounding effects of environment resets or simulator feedback.
>
> **W2:** We agree with the reviewer that dynamic obstacles are critical for real-world deployments. The datasets we use for training and evaluation, such as SCAND and ReCon, **already contain** moving pedestrians, moving objects, social interactions, and other dynamic elements. The reported metrics therefore **already reflect performance in both static and dynamic settings**, and UniWM remains competitive with state-of-the-art baselines. To strengthen this aspect, in the revised version we will add visual demonstrations of dynamic-obstacle scenes from SCAND and ReCon, making explicit UniWM’s ability to handle dynamic environments. We thank the reviewer for pointing out this important clarification.
>
> **W3 & Q3:** We acknowledge the reviewer’s concern regarding efficiency and inference speed. To address this, we provide a direct comparison of average inference time per trajectory together with navigation metrics (SR, ATE, RPE) across the four datasets (Go Stanford, ReCon, SCAND, HuRoN).
>
> | Method                  | Avg SR | Avg ATE | Avg RPE | Avg Inference Time per Trajectory (second) |
> |--------------------------|--------|---------|---------|-----------------------------------|
> | NoMaD (Sridhar et al.24) | 0.48   | 0.79    | 0.26    | **0.6**                              |
> | NWM (Bar et al.25)       | 0.55   | 0.63    | 0.23    | 114✖️32                             |
> | Anole-7B (Chern et al.24)| 0.27   | 1.80    | 0.73    | 654                              |
> | UniWM (with Memory)      | **0.78**   | **0.32**    | **0.12**    | 82                               |
> | UniWM + Quant. 4-bit (with Memory)      | -  | -   | -  | 16 (est. [6])                               |
>
> As shown in Table above, NoMaD achieves fast inference but lacks imagination capability, which limits success rates in challenging cases. World‑model based approaches (NWM, Anole‑7B, UniWM) incur higher inference costs due to visual imagination. Importantly, UniWM runs substantially faster than its backbone Anole‑7B and NWM, while delivering markedly better navigation performance, demonstrating **a favorable balance between efficiency and accuracy**. Quantization to 4-bit [6] can potentially power UniWM up to average 16s per trajectory.
>
> Note that for NWM, we follow its best reported configuration and train from scratch on all four datasets. Because NWM uses an MPC-style strategy that ranks 32 candidate trajectories and simulates each one to select the best, we report its inference time as “114 × 32 seconds” to reflect the actual average cost per executed trajectory. We will include this discussion and table in the revised manuscript.
>
> **Q1:** We selected Anole-7B because it provides a unified multimodal Transformer architecture with strong joint modeling of text and vision tokens, and crucially **supports direct prediction of image tokens**. Compared to alternatives like Qwen-VL-2.5, which cannot directly predict image tokens, Anole-7B offered better compatibility with our bin-token discretization scheme and reconstruction objectives, enabling stable training across both planner and world-model roles. This made Anole-7B the suitable backbone for UniWM’s unified training strategy.
>
> **Q2:** We appreciate the reviewer’s suggestion to include a comparison with Aether [7]. Table below reports results for Aether and UniWM. Since the official Aether repository does not provide training code, we were unable to train Aether from scratch on our datasets. Instead, we used the released checkpoint together with its goal‑conditioned visual planning script for evaluation. Under this setting, **UniWM achieves clearly stronger performance against Aether**. The results are reported below:
>
> | Method                  | SR ↑ | ATE ↓ | RPE ↓ | SSIM ↑ | LPIPS ↓ | DreamSim ↓ | SSIM@5 ↑ | LPIPS@5 ↓ | DreamSim@5 ↓ |
> |--------------------------|------|-------|-------|--------|---------|------------|----------|-----------|--------------|
> | Aether [7]   |  0.32   |   1.37   |   0.51   |   0.373    |   0.271     |     0.068      |   0.235     |     0.514     |      0.158       |
> | UniWM | **0.71** | **0.36**  | **0.13**  | **0.457**  | **0.254**   | **0.041**      | **0.350**    | **0.435**     | **0.126**        |

---

> > ### Author Response · Authors · 2025-11-20
> >
> > **Reference**
> >
> > [1] Zhang, Kaiwen, et al. "Epona: Autoregressive Diffusion World Model for Autonomous Driving." arXiv preprint arXiv:2506.24113 (2025).
> >
> > [2] Garg, Anant, and K. Madhava Krishna. "Imagine-2-drive: High-fidelity world modeling in carla for autonomous vehicles." arXiv e-prints (2024): arXiv-2411.
> >
> > [3] Chi, Xiaowei, et al. "MinD: Unified Visual Imagination and Control via Hierarchical World Models." arXiv preprint arXiv:2506.18897 (2025).
> >
> > [4] He, Haoran, et al. "Learning an actionable discrete diffusion policy via large-scale actionless video pre-training." Advances in Neural Information Processing Systems 37 (2024): 31124-31153.
> >
> > [5] Alonso, Eloi, et al. "Diffusion for world modeling: Visual details matter in atari." Advances in Neural Information Processing Systems 37 (2024): 58757-58791.
> >
> > [6] Frantar, Elias, Saleh Ashkboos, Torsten Hoefler, and Dan Alistarh. "Gptq: Accurate post-training quantization for generative pre-trained transformers." arXiv preprint arXiv:2210.17323 (2022).
> >
> > [7] Zhu, Haoyi, et al. "Aether: Geometric-aware unified world modeling." Proceedings of the IEEE/CVF International Conference on Computer Vision. 2025.

---

> > ### Comment · Reviewer_zu2f · 2025-11-28
> >
> > Thank you for the authors’ response, which addresses my concerns regarding the world model’s prediction quality, the rationale for adopting Anole-7B as the backbone, and the dynamic obstacle avoidance capability. However, my concern about inference speed persists. As indicated in the table, UniWM requires 16 seconds for a single trajectory planning iteration—this would render real-world deployment extremely inefficient. Could the authors propose alternative approaches to achieve real-time path planning within the current framework?

---

> ### Author Response · Authors · 2025-11-29
>
> We are happy to have **already addressed your concerns** regarding the world model’s prediction quality, the rationale for adopting Anole‑7B as the backbone, and the dynamic obstacle avoidance capability. We sincerely appreciate your acknowledgement of these clarifications. Your remaining concern about inference speed is well‑taken. While UniWM is currently estimated to require ~16 seconds per trajectory under 4‑bit quantization, we view this as **an important optimization frontier**, as the time cost of repeatedly leveraging VLMs to predict high‑dimensional image tokens is indeed non‑negligible. At the same time, we emphasize the **efficiency–accuracy tradeoff already achieved by UniWM**. As shown in Table above, NoMaD achieves fast inference but lacks imagination capability, which limits success rates in challenging cases. World‑model based approaches (NWM, Anole‑7B, UniWM) incur higher inference costs due to visual imagination. Importantly, UniWM runs substantially faster than its backbone Anole‑7B and NWM, while delivering markedly better navigation performance, demonstrating a **favorable balance between efficiency and accuracy**. On the other hand, **we are actively exploring strategies** such as model compression and distillation, transferring UniWM’s reasoning ability into lighter student models, and parallelized denoising with hardware acceleration to reduce per‑step latency. These directions aim to accelerate the navigation process and **move toward real‑time path planning within the current framework**.
>
> We emphasize that our main contribution lies in introducing **the first unified, memory‑augmented world model paradigm** for goal‑conditioned visual navigation, integrating egocentric visual foresight and planning within a single multimodal autoregressive backbone. A hierarchical memory mechanism further integrates detailed short‑term perceptual cues with longer‑term trajectory context, while efficiency optimization is a **natural next step** toward real‑time deployment.

---

### Official Review · Reviewer_Vagp · 2025-11-01

**Soundness:** 3
**Presentation:** 3
**Contribution:** 4
**Rating:** 6
**Confidence:** 2

**Summary:**

Paper proposes a novel unified, memory-augmented world model, UniWM, which integrates egocentric "visual imagination" and route planning within a single model. The unified design allows the information to be tightly integrated between the 2 tasks (prediction and control). A hierarchical memoty module combines the sensor embeddings with the world context for long-term planning. In 4 benchmarks, the proposed UniWM improves success rates, and reduces trajectory errors. The zero-shot setup in the unseen TartanDrive also shows the generalizability of the proposed method.

**Strengths:**

1. Method proposes a novel architecture which significantly outperform existing SOTA for 4 benchmarks, and a zero-shot setting.

2. Quality of paper is high in term of architecture design and experimental results.

3. Paper is clear, but requires some background knowledge of the field.

4. Paper is highly significant in the significant improvements (up to 30%) for the ego-centricity navigation task.

**Weaknesses:**

Paper highlights domain shift and fixed token budget as 2 limitations.

For the domain shift issue, the proposed method does not seems to have a good potential solution.

**Questions:**

1. Paper highlights domain shift and fixed token budget as 2 limitations. But only proposes some possible solutions for Fixed token budget, and not domain shift. Please suggest some possible solutions/mitigation methods for the domain shift issue.

---

> ### Author Response · Authors · 2025-11-20
>
> **W1 & W2 & Q1:** Thank you for raising this point! While the paper highlights domain shift as a limitation, we would like to clarify that several design choices in UniWM already aim to mitigate this issue. Specifically, the **unified training scheme** enables action decisions to be grounded in imagined visual outcomes, which reduces state–action misalignment when the environment distribution changes. In addition, the **hierarchical memory mechanism** stabilizes rollouts by combining immediate perceptual cues with longer‑range trajectory context, which improves robustness in unseen domains. These elements explain why UniWM achieves **competitive zero‑shot generalization** on TartanDrive without fine‑tuning (Sec. 3, Table 6).
>
> Beyond what is already implemented, there are several practical directions to further address domain shift:
>
> 1) Test‑time adaptation and uncertainty gating. During inference, the model can update a subset of parameters online using unsupervised objectives. Coupling this with uncertainty estimation enables the agent to fall back to conservative actions when predictions are unreliable.
>
> 2) Targeted data augmentation. Introducing synthetic ego‑robot artifacts, varying camera intrinsics, or applying domain randomization (lighting, textures, motion blur) during training can reduce sensitivity to appearance differences that often drive domain shift.
>
> 3) Representation alignment. Employing more robust pretrained encoders or adding feature‑level alignment losses can make latent representations less dependent on low‑level distributional details.
>
> In the revised manuscript, we plan to add a short discussion of these mitigation strategies. We believe these additions will directly address the reviewer’s concern and strengthen the paper’s contribution.

---

### Meta-Review · Area_Chair_YHxt · 2026-01-07

**Summary:**

Reviewers generally find the paper clearly written and the results strong. The main contribution is a unified planner/imagination model with a hierarchical KV-cache memory. That said, the discussion converged on several decision-critical issues: the unified action+future-image setup feels incremental, goal-conditioning the transition model raises methodological concerns, some baseline comparisons were questioned for fairness, and the method appears too slow for practical deployment.

The rebuttal improves the empirical credibility by clarifying that NWM was retrained using the official codebase and NoMaD preprocessing, and by adding a separate-vs-unified comparison that supports the "unification helps" narrative. Still, core concerns remain: one reviewer continues to view goal-conditioned transitions as conceptually problematic, and another emphasizes that 16s per planning iteration is far from usable in real settings. Given the remaining conceptual objection, latency, and only moderate novelty, I lean Reject.

**Reviewer Concerns:**

Vagp: The main concern is robustness under domain shift and whether the method generalizes beyond the tested settings.

zu2f: The main concern is deployability, as the inference and planning cost appears too high for practical use.

fGwF: The main concern is limited novelty and whether the comparison to NWM is fully fair and reproducible.

LfkC: The main concern is that goal-conditioning the transition model is conceptually problematic and may bias predictions.

**Reviewer Scores:**

Vagp: 6

zu2f: 6

fGwF: 4

LfkC: 4

---

### Decision · Program_Chairs · 2026-01-26

Reject